# Advancing the representation of reservoir hydropower in energy systems modelling: The case of Zambesi River Basin

Nicolò Stevanato[1,2]*, Matteo V. Rocco[1], Matteo Giuliani[3], Andrea Castelletti[3], Emanuela Colombo[1]

1 Department of Energy, Politecnico di Milano, Milan, Italy, 2 Fondazione Eni Enrico Mattei, Milan, Italy, 3 Department of Electronics, Information, and Bioengineering, Politecnico di Milano, Milan, Italy

* nicolo.stevanato@polimi.it

**Data Availability Statement:** All data files and code scripts are available from the Zenodo repository https://doi.org/10.5281/zenodo.5242926.

## Abstract

In state-of-the-art energy systems modelling, reservoir hydropower is represented as any other thermal power plant: energy production is constrained by the plant's installed capacity and a capacity factor calibrated on the energy produced in previous years. Natural water resource variability across different temporal scales and the subsequent filtering effect of water storage mass balances are not accounted for, leading to biased optimal power dispatch strategies. In this work, we aim at introducing a novelty in the field by advancing the representation of reservoir hydropower generation in energy systems modelling by explicitly including the most relevant hydrological constraints, such as time-dependent water availability, hydraulic head, evaporation losses, and cascade releases. This advanced characterization is implemented in an open-source energy modelling framework. The improved model is then demonstrated on the Zambezi River Basin in the South Africa Power Pool. The basin has an estimated hydropower potential of 20,000 megawatts (MW) of which about 5,000 MW has been already developed. Results show a better alignment of electricity production with observed data, with a reduction of estimated hydropower production up to 35% with respect to the baseline Calliope implementation. These improvements are useful to support hydropower management and planning capacity expansion in countries richly endowed with water resource or that are already strongly relying on hydropower for electricity production.

## 1. Introduction

### 1.1. Energy and water

Water [1] and Energy [2] are recognized by the United Nations as two of the 17 Sustainable Development Goals (SDGs) that humanity should pursue before 2030 for achieving sustainable development [3]. Water is a basic human right [4], no society can survive and prosper without it. On the other hand, energy is an instrumental human right: energy itself does not determine human dignity but with zero or poor access, fundamental rights may not be

**Funding:** MG and AC were supported by DAFNE-Decision Analytic Framework to explore the water-energy-food Nexus in complex transboundary water resource systems of fast developing countries research project funded by the Horizon 2020 programme WATER 2015 of the European Union, GA 690268. Data from the mentioned project were used. There was no additional external funding received for this study.

**Competing interests:** The authors have declared that no competing interests exist.

guaranteed [5]. Energy and water challenges are not independent and their interconnections, often entitled *Water-Energy nexus* [6, 7], are increasingly recognized and studied [8–12]. "Water is needed for each stage of energy production, and energy is crucial for the provision and treatment of water" [7], and with the increase of needs for both energy and water worldwide, scientifically solid policies that regulate the energy sector and its water use and withdrawal without hindering energy security are needed to prevent future stress risk in particularly vulnerable areas.

Hydropower is the largest renewable source of electricity generation worldwide, accounting for 16% of the global electricity generation mix and 61% of the renewables [13], and a key component of the water-energy nexus. Hydropower generation is largely dependent upon climate variability, which may either curtail production during intense drought events that reduce water availability [14] or induce large water losses as a consequence of spilling during flood events [15]. In addition, hydropower production is also influenced by other competing water users such as irrigation, domestic and industrial water supply, and ecosystem preservation (e.g., 40% of existing hydropower dams serve multiple demands [16]). Despite the dynamic and non-linear nature of hydropower in phenomena such as evaporation losses and hydraulic head variation patterns is well understood and considered in hydrological studies, with few exceptions, energy system models fall short in considering the holistic influence of hydrometeorological variability on bulk power systems [17]. Indeed, energy systems models traditionally work under the assumption of hydrological stationarity [18], neglecting water resource variability and uncertainty at the core of hydrological models. On the other hand, in many state-of-the-art hydrological models (e.g. MIKE [19], RIBASIM [20], RiverWare [21], WEAP [22], Pywr [23], SWAT [24] among the others) the energy system is not comprehensively included in the modelling scope and hydropower electricity dispatch is usually defined without accounting for energy grid constraints.

## 1.2. Modelling hydrological and energy systems

In the identified context, different efforts have been carried out in the past to face the issue of properly representing hydrological and energy systems in modelling frameworks, to achieve more scientifically solid energy and hydrological planning, strictly interdependent one from the other. In some cases, two existing models (e.g. an energy model and a hydrological model) were integrated, with different levels of integration, to exploit the unique characteristics of both the frameworks, or in other cases, new models were created from scratch to perform energy and hydrological planning in the study areas.

For example, if two models are used without integration, the interdependency between energy and water systems is indirectly considered through the input variables employed in each one of these two models. This approach is adopted by Voisin et al. [25], coupling a water model with an electricity production cost model to simulate energy generation and power dispatch. A similar approach is adopted by De Vita et al. [14] where externally calculated hydrological data are fed to the energy model PRIMES. The advantage of this approach lies in its low computational cost, making it suitable to be applied to large scale systems. On the other hand, disadvantages are in the absence of a feedback dynamic between the two models, being it either soft- or hard- link, that may return conflicting results.

Soft linking the two models is an option in which the energy and water models are still separate, each based on its own objective functions and data structures but working cooperatively in a loop exchanging parameters and converging to a unique solution expressing the trade-off among models' objectives. This approach was applied in Fernandez-Blanco et al. [26], who developed a linear programming model able to describe the hydropower component and the

power dispatch with dependencies on reservoirs' releases for plant cooling. Other examples of soft integration are represented by Pereira-Cardenal et al. [27], Chowdhury et al. [28], who make use of the hydrological model VIC-Res to estimate the available hydropower at each dam in a system, and insert such information in the model PowNet to determine the hourly dispatch of electricity, and by Agrawal et al. [29], whom soft linked the energy model LEAP [30] and the water model WEAP. Advantages of this method consist in the fact that pre-existing models can be employed since the water and energy systems models are kept separate. However, it is not always easy to identify how to link the objective functions of the two different models. In addition to that, this approach is significant if the two models come to convergence, and it is not easy to predict whether this approach will lead to such a result and the related computational effort.

When the two models are integrated into a single model, the joint optimization will consider, according to one unique objective function, how to allocate the resources in the two systems and how to balance the trade-offs among the two native models. Costs and benefits can be easily quantified for an energy system, but this is not the case for water systems. Natural water is often unvalued or undervalued: the real economic value of water is hardly accurately quantifiable, as well as the costs and benefits deriving from its marginal usage. On the other hand, the same can be more easily done for energy. Examples of this are found in Payet-Burin et al. [31]—who evaluate the connection among overall energy production and cooling water for thermal plants in the Iberian Peninsula according to a fully integrated model -, Khan et al. [32] built a fully coupled water-energy optimization model which hard-links the two systems in detail across spatial and temporal scales, and Su et al. [17] whom create an ad-hoc model for the United States' West Coast electricity and hydrological systems. The advantage of a fully integrated model resides in the capability to assess the interactions among energy and water systems without approximations in their dynamic behavior; however, this is balanced by the need to merge, harmonize and run simultaneously two models usually defined with different time and space scopes and with the level of detail required by the modelled complex phenomena. At present, a model where water resources management and energy system planning and management are optimized simultaneously has not yet been implemented because of its high computational cost.

Finally, It is indeed worth mentioning planning techniques like the one represented by Gonzalez et al. [33], who developed a model from scratch that takes into account both energy and water systems, their interconnections and performs multi-objective optimization over a horizon of 50 years, taking into account the inherent uncertainty to plan multi-purpose reservoir systems. At the same time, Sterl et al. [34] developed a new framework for assessing intra-regional benefits of using reservoir hydropower as a balancing method for grids with high penetration of renewables in the West African Power Pool (WAPP).

The issue with the presented cases is the lack of replicability of those integrated energy-water frameworks. The mentioned approaches are indeed rarely replicated to other geographical areas, given the complex structure of the integrated models, which are sometimes ad-hoc developed around a specific case study. The scientific value of such integrated models is indeed relevant and not questioned in this work, which is instead focused on model replicability in broader engineering applications.

On the other hand, most energy system models (both proprietary and open-source) are adopted as a standard by the international community [35] and used in a wide range of applications, as discussed later in section 'Aim of the work'. However, their representation of a variety of engineering phenomena, including hydropower, leaves space for improvements. Therefore, improving an already existing energy modelling framework may have a broader

impact on the international community in terms of model accessibility by users and replicability of the analysis in other contexts.

Given these considerations, this work proposes a methodology for improving the reservoir hydropower representation in open-source energy system models.

## 1.3. Representation of hydropower in energy modelling

A non-comprehensive review of the literature concerning hydropower representation in energy system modelling is carried out with the sole scope of highlighting criticalities in current approaches in the energy modelling science. The analysis considers the most adopted and known, both proprietary and open-source, energy modelling frameworks.

Depending on the purpose of the analysis, *energy systems models* provide a technical description of energy dispatch and/or power capacity expansion by including one or multiple energy carriers and by relying on different time resolutions and spatial aggregation. Established models of this kind are MARKAL [36], which then evolved into TIMES [37]–known for being adopted by the International Energy Agency (IEA) in their modelling programme ETSAP [38]–MESSAGE [39]–developed by the International Institute for Applied System Analysis (IIASA) and a version of which is adopted by the International Renewable Energy Agency (IRENA) and International Atomic Energy Agency (IAEA) in their analysis–and LEAP [30], a very long standing energy system model, used, among others, by the United Nations Framework Convention on Climate Change (UNFCCC). Emerging challenges in energy modelling include: (I) modelling high penetration of renewable energy sources with high temporal variability and different types of storage technologies [40]; (II) modelling a highly spatially decentralized energy production system [41]; (III) verifying modelling assumptions, checking and updating background data and examining models' structures [42]. Several Open Source modelling frameworks have recently emerged to address the above-mentioned challenges, such as OSeMOSYS [43], Balmorel [44], PyPSA [45] and Calliope [46], and many others revised in the most recent reviews on the theme [47, 48] and listed in the Open Energy Modelling Initiative Wiki [49] as suggested in [50].

Characterization of hydropower is a major weakness of energy systems models, mostly due to (i) the fact that all models deal with linear programming optimization problems, and reservoir hydropower is characterized by non-linear stochastic phenomena that need to be simplified to be included in such models and (ii) to the strong discrepancy between the characteristic time of hydro and power phenomena. Typically, widely adopted only-energy models, both proprietary and open-source, deal with hydropower just by exogenously constraining its production based on the installed capacity and on a so-called capacity factor calibrated based on historical energy production data [51]. An example of the use of TIMES for specific hydropower modelling can be found in Carvajal et al. [52] where the model is used to represent the highly hydropower-dependent national electricity system of Ecuador, and the authors clearly state how "different availability factor definitions are used to represent electricity dispatch for [. . .] flexible reservoir systems". An example of the use of MESSAGE explicitly including hydropower constraints is represented by Fairuz et al. [53], where a capacity factor of 24% and 45% is indicated for "Hydro Storage" and "Run of River" technologies respectively in the analysis of long term scenarios for the Malaysian electricity mix. Another example is represented by IRENA in [54], in which they model the Southern African Power Pool (SAPP) exploiting the energy model SPLAT, based on MESSAGE. In this work, IRENA evaluates the most affordable investment required by the power generation sector also in terms of domestic and international transmission to meet SAPP growing energy demand, in this work hydropower is described simply providing the expected energy production in a year without accounting for water resource availability.

One of the main benefits of open-source software is the possibility for the users to modify, improve and adapt such models in order to meet their specific requirements and needs. Some examples of existing open-source energy system models, their representation of hydropower, and some efforts carried out to better represent it are reported below. A first example is provided by the experience of OSeMOSYS, one of the most long-standing open-source energy models. In OSeMOSYS hydropower technology is modelled only through a capacity factor, which can be differentiated for different time slices. In a "fix-and-relax" version of OSeMOSYS, mentioned in [55] and available at [56], the model is enhanced under the hydropower representation point of view. Balmorel considers reservoir hydropower as a supply technology with integrated storage, to store energy available but not supplied [57]. A good example of how PyPSA deals with the matter is in Schlott et al., their PyPSA Vietnam model [58] makes use of a potential energy approach for modelling both run-of-river and reservoir hydropower based on water runoff data and then normalized on current hydropower hourly generation. Finally, the energy modelling framework Calliope is characterized by the possibility to customize every energy conversion technology, leaving maximum freedom to the user to define the technologies involved according to specific needs. In this context, hydropower can be characterized through exogenous inputs, like historical data of production of existing power plants, or capacity factors. Good [59] attempts to improve the Calliope model with a single water reservoir system, investigating the effects of a daily to seasonal balancing of the Swiss energy grid by providing the model with the real operation timeseries of plants in terms of energy produced; in presence of a dam, a battery energy storage technology was added to allow the model to store hydroelectric energy in the reservoir.

## 1.4. Aim of the work

As observed in chapters 1.2 and 1.3, model integration between energy and hydro models results in drawbacks in terms of computational time and non-replicability, among others. On the other hand, while only-energy models, and in particular open-source ones, are meant to build in the direction of solving such issues, they lack proper characterization of hydrological phenomena and hydropower production. For these reasons, the aim of this work is to provide a methodology for improving the characterization of reservoir hydropower in open-energy-modelling. Through this methodology, we aim at reaching an advance in the energy modelling science, by learning from the integrated modelling experience analyzed in chapter 1.2, and overcoming the highlighted limits thanks to the advantages brought about by the open-modelling framework. Building on the experience of the open energy modelling community [60], and the observed benefits of "communities of practice" that concentrate efforts in improving existing open-source energy models [55], the aim of this work is to contribute to improving the representation of reservoir hydropower within open-source energy systems modelling by implementing physical constraints of multi-reservoir hydropower systems, bringing novelty in the field of hydropower energy modelling. Our approach does not aim at integrating two different models but improving the way reservoir hydropower is represented in existing energy system models. Among the variety of existing open energy modelling frameworks [49], Calliope [46] is selected due to its bottom-up technology-driven approach and full customizability of the involved energy conversion technologies [61], which makes it an ideal ground for testing the proposed approach. Furthermore, the framework is widely adopted for the study of national power systems [62–64] and interconnected power pools [65–68] in literature. From the observed past works outlined in Section 1.2 we derive the most important lessons learned in the analyzed approaches, and we propose an improvement of reservoir hydropower representation in energy modelling. With respect to previous representations of reservoir

hydropower in energy system models, outlined in Section 1.3, our approach introduces the following novelties:

- Multiple cascade reservoir systems including inflow patterns, maximum and minimum storage limits, and maximum water release constraints can be modelled and integrated with the overall energy system.

- Non-linear hydrological constraints such as evaporation losses and time-variable hydraulic head are modelled based on an external computation loop to keep the integrated model linear.

- The space scope of the model can be extended to multiple countries and by including multiple sub-regions in order to encompass the cascade of multiple basins and to capture the local availability of natural resources.

- The *Calliope* open-source linear optimization energy modelling framework [46] allows modelling the operation of energy systems with different space scopes and high physical detail, including multiple energy carriers and hourly time resolution.

Our water-enhanced energy model is demonstrated in the case study of the *Zambezi River Basin* as part of the overall *Southern African Power Pool* (SAPP) and comparatively analyzed with a traditional Calliope model adopting a traditional description of hydropower generation. In the SAPP, many countries are already strongly dependent on hydropower plants of the Zambezi River Basin, both for domestic production and export. Moreover, nearly 40 GW of hydropower could be potentially deployed in this region in the short to medium term to meet growing energy demands [69]. For such reasons, hydropower is expected to play a key role in the future power generation mix as well as at present, and more realistic and accurate modelling of the hydropower source is therefore essential.

## 2. Methods and models

A detailed hydropower plant operational model should consider the dynamic availability of water resource, which depends on the following parameters: (I) time-dependent water inflow patterns in the basin due to hydrological processes; (II) water inflow supplied by upstream reservoirs (i.e. reservoirs linked in multiple cascades); (III) dam maximum and minimum operational level and related water release constraints (in terms of maximum amount of water processable); (IV) evaporation losses dependent upon the reservoir surface. All these parameters are ultimately affecting the reservoir dynamic hydraulic head and the power that can be produced by reservoir hydropower.

The proposed modelling strategy starts from the basic formulation of the *Calliope* model (for the purpose of this work called: *Calliope_Base*) (https://calliope.readthedocs.io/en/stable/), consisting of an optimization model for the analysis of the operation of energy systems of different scales with high physical details and hourly time resolution, based on the power nodes modelling approach [70]. The Calliope model is enhanced by modelling the physical constraints that deal with the four parameters related to hydropower and water basins management previously listed (the model resulting from the implementation of the proposed intervention will be called: *Calliope_Hydro*).

The network of *multiple cascade water reservoirs* is modelled by linking multiple elements of the, already existing in Calliope_Base, *storage* technology: specifically, each water storage includes a description of the maximum water release and provides the real water availability through exogenously provided inflow patterns, depending on precipitations and upstream inflows (see sub-section 2.1 for a detailed description). Once the reference energy system is

characterized, including the above-mentioned cascade reservoirs, the model returns the electricity dispatch and the hourly time series for the storage water level over the analyzed time horizon.

The hourly power dispatch profile (i.e. the decision variable of the optimization problem) is non-linearly linked to the hydraulic head of the reservoirs, affected by the evaporation losses, and influencing the power production by hydropower plants. To preserve the model linearity, an exogenous numerical loop is defined to correct the time series for reservoirs hydraulic head to account for the effects of such *storage dependent variables*, which ends when the iterative problem reaches the numerical convergence (see sub-section 2.2 for a detailed description).

Sections 2.1 and 2.2 describe the modelling methodology adopted for enhancing *Calliope_-Base* into *Calliope_Hydro*, taking into consideration the four points highlighted at the beginning of this chapter. Section 3 presents the case study and how the new methodology is applied to it, while section 4 presents and comments the results.

## 2.1. Modelling multiple cascade water reservoirs

In the current version of Calliope, here referred to as *Calliope_Base*, the modeler can make use of a series of predefined supply and conversion technologies archetypes (i.e. *Supply*, *Supply +*, *Conversion*, *Conversion +*, *Storage*) and the standard way of modelling reservoir hydropower is employing a Supply + technology, a technology that allows a flow of energy to enter the system based on specific resource availability, and that can have storage integrated, (like for example a PV power plant with a Battery Energy Storage System). Such existing technology archetypes are used here to better model reservoir hydropower.

In this subsection, we outline how we dealt with points (I) time-dependent water inflow patterns in the basin due to hydrological processes and (II) water inflow supplied by upstream reservoirs.

The strategy for defining a network of multiple water reservoirs within a reference energy system in Calliope is schematically provided by Fig 1 for a system composed of 2 reservoirs and conceptually extendable to an n-reservoirs system.

With reference to Fig 1, each water reservoir is modelled based on the already existing *storage* technology S#, characterized by: storage capacity, depth of discharge (defining the minimum operational level at which water can flow through turbines), storage level at the initial and time condition (including the possibility to force as well final conditions or define cycling storage where initial and final conditions coincide), and maximum water release (i.e. the maximum water flow rate that can be turbined).

The inflows into the reservoir due to precipitations and tributary rivers (excluding water released by upstream reservoirs) are represented as the already existent *supply* technology SW#, providing exogenous timeseries of water inflows.

The energy conversion technology, consisting in the hydraulic turbine, is represented by the already existing conversion technology named *HP#*, and it converts the energy embedded in the hydraulic head $H[m]$ of a given amount of water flow rate $\dot{m}_w[kg/s]$ into electrical power $\dot{E}_e[W]$ as represented by Eq (1), depending on the efficiency of the turbine $\eta_e[-]$ and hydraulic head of the system $H[m]$ (notice that $g = 9.8\ m/s^2$ is the gravity acceleration). The hydraulic head is computed as the difference between current water level in the dam and the elevation of the turbine axis. The tailwater effect is not considered.

$$\dot{E}_e = \dot{m}_w \cdot g \cdot H \cdot \eta_e\ \ [W] \tag{1}$$

Finally, the *conversion plus* technology C+ is designed to take all the water released by a generic reservoir that flows through turbines to the downstream reservoirs. Notably,

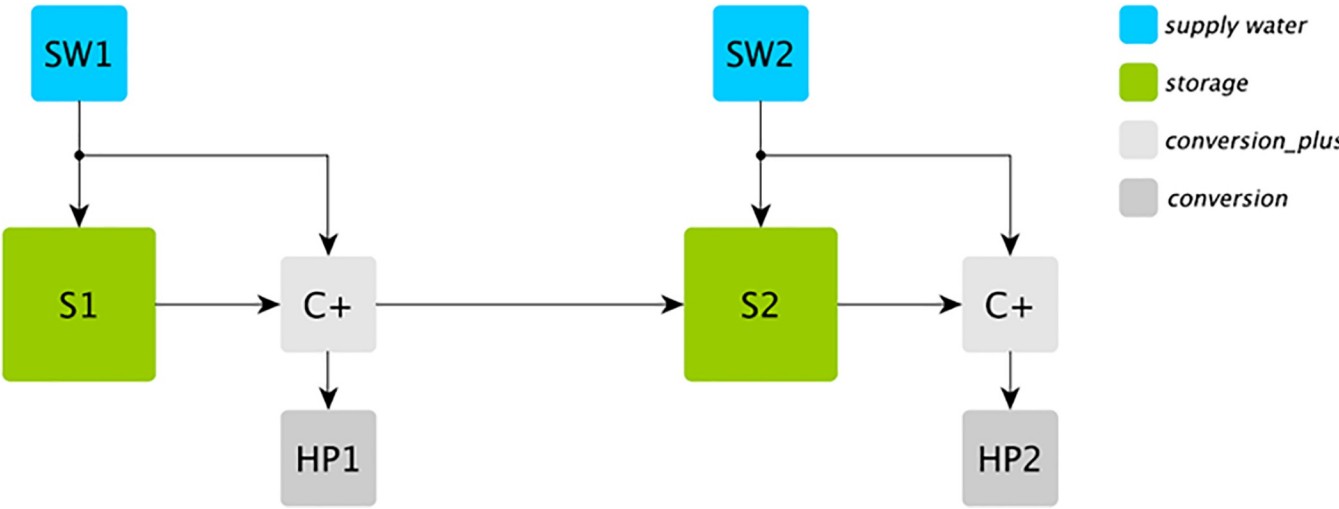

**Fig 1. Example of a multiple cascade water reservoirs within the Calliope reference energy system.**

with reference to Fig 1, the C+ block possibility of having multiple outputs allows us to *duplicate* the water release, while the first output stream is supplied to the hydropower plant (HP) to produce electricity, the twin-stream flows into the following cascade reservoir. The C+ block allows then to connect two reservoirs in cascade operating in different locations through a transmission line mimicking water flows crossing different locations.

## 2.2. Modelling storage dependent variables

In this subsection, we outline how we dealt with points (III) dam maximum and minimum operational level and related water release constraints and (IV) evaporation losses dependent upon the reservoir surface.

To achieve a full representation of hydrological dynamics within the energy model, the links between water evaporation losses and hydropower production with the hydraulic head of the reservoirs (i.e. the water storage level) must be clearly defined. Due to the linear nature of the Calliope model, there is no way to define dependencies between variables at each time step of the simulation, neither it would be possible to further develop such a feature without making the model non-linear. For such reasons, in this work evaporation losses and hydraulic head are iteratively changed within an external loop updating the Calliope water storage time series until the convergence is reached (Figs 2 and 3). This approach was adopted in previous work by Del Pero et al., who applied this logic to deal with non-linearities in energy modelling of a smart district in Italy [71].

The *evaporation* of water stored in a dam could be an important source of losses, especially for very extended reservoirs in tropical climate zones. Calliope provides a predefined constraint that allows defining a storage loss as a fraction of total capacity per hour (also in the form of timeseries). In this way, storage losses due to evaporation are evaluated at each time step and passed to the model as an input. First, evaporation losses $ev_{loss,t}[m^3/h]$ are evaluated for each time step $t$ based on Eq (2) as a function of a net evaporation coefficient $ev_{coeff,t}[m/h]$ (itself function of storage level and of the time) and of the reservoir surface $S_t[m^2]$ (function of the reservoir shape and the time). Then, the storage losses timeseries $s_{loss,t}[1/h]$ are computed

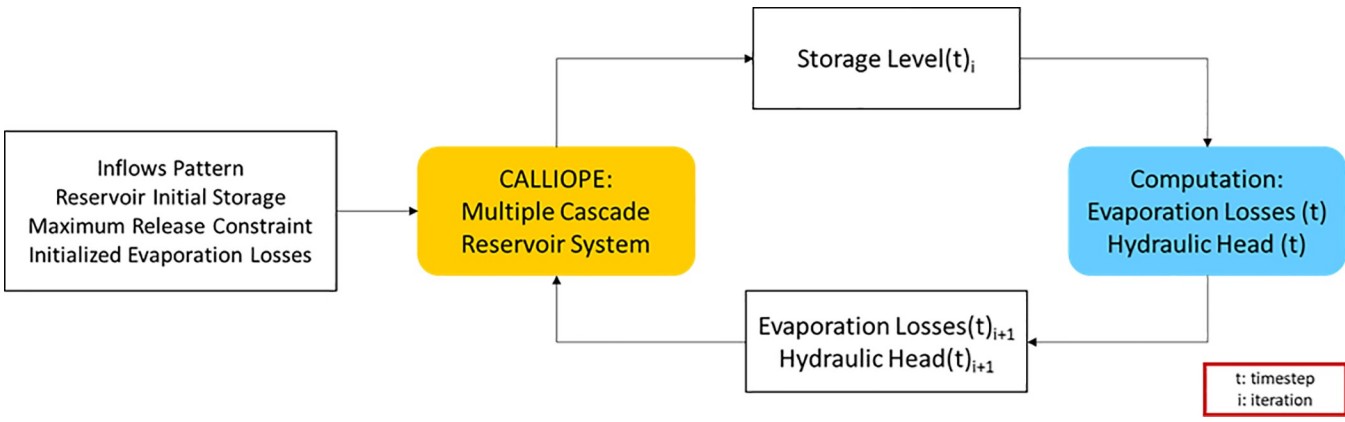

**Fig 2. Logical scheme for the iteration of the *Calliope_Hydro* model.**

as in Eq (3) as a function of evaporation losses and the storage overall capacity $s_{cap}[m^2]$.

$$ev_{loss,t} = ev_{coeff,t} \cdot S_t \tag{2}$$

$$s_{loss,t} = \frac{ev_{loss,t}}{s_{cap}} \tag{3}$$

If the change in storage level is of the same order of magnitude as the overall hydraulic head, the latter parameter may change significantly, strongly affecting the hydroelectric power $\dot{E}_e[W]$ (see Eq (1)).

Since the storage level and the hydraulic head in every time step are results of the model, the reservoir surface at each time step can be easily computed once its *storage-surface* curve is known. Once the timeseries of the evaporation losses and the initial values of the time-varying hydraulic head are known, it is possible to pass these parameters exogenously to the model in a second run of the optimization. This second run of the optimization will result in a new time-series of storage levels and hydroelectric power output. This process is repeated until the storage timeseries reach convergence.

## 2.3. Current limitations

Among the observed methodologies in previous works, two main phenomena are left out of the proposed work. The main shortcomings of the proposed approach are as follow:

- Spillage: due to the linear nature of the optimization framework of Calliope, it was not possible to accurately represent the activation of the dam spillways, i.e. releasing water out of the reservoir without whirling it in cases of excess water to avoid dangerous overcharge of the dams. To better reproduce water spillages, it would be necessary to define a Boolean constraint dependent on the volume of water stored in the reservoir, which is activated when the storage exceeds a certain threshold. The proposed *Calliope_Hydro* instead models the spillage as a conversion technology connecting two cascade reservoirs with a conversion efficiency equal to zero. We made this choice to discourage the arbitrary allocation of water from upstream to downstream reservoir according to an economical optimization. Setting the efficiency equal to zero, the spillage is modelled as a wasted water flow in order to be minimized by Calliope optimization.

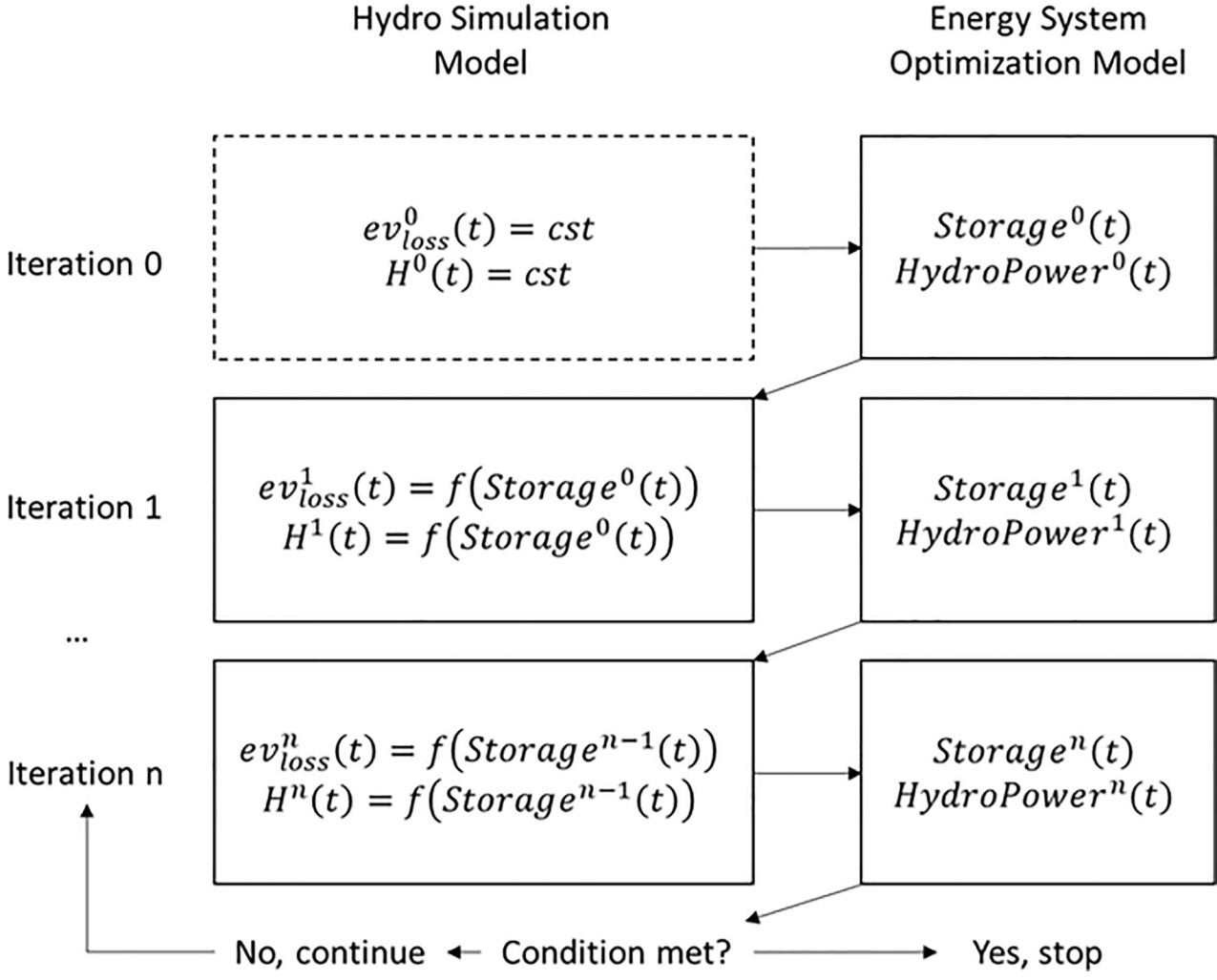

**Fig 3. Logical scheme of iteration between the two models.**

- Multi-Stakeholder Management, as mentioned in the introduction of this work, is another critical aspect of water resource management, and consists in enlarging the scope of the analysis to include other water-related sectors beyond power production needs. This would require the formulation of other *water demands* alongside the electricity one, to account for competing uses of water in the river basin. Not considering the non-hydropower uses of water implies that the amount of water that appears to be available for the turbines according to the model is actually more than the amount available in real life. The main other use of water in the area is water for agriculture, which is extracted directly from the river basin and the dams, the water used for agriculture should hence be subtracted from the amount of water flowing in the system. This could be modelled in the framework by inserting, as stated above, a specific demand for *non-hydropower water*, that has to be met by the system; that water would then be removed from the system.

These improvements to overcome current limitations are planned to be integrated in future work.

## 3. Case study

Africa is home to five regional power pools: Maghreb Electricity Committee (COMELEC), Eastern Africa Power Pool (EAPP); Central African Power Pool (CAPP); West African Power Pool (WAPP); and Southern African Power Pool (SAPP) [72]. SAPP includes the Democratic Republic of the Congo, Tanzania, Angola, Zambia, Malawi, Namibia, Botswana, Zimbabwe, Mozambique, Lesotho, South Africa, and eSwatini. The overall installed power capacity is 50 GW, with a yearly generation of 400 TWh of electricity, and with a regional demand expected to increase 4.5 times by 2040 up to 1000 TWh [73]. Despite coal-fired power plants dominate the regional power mix, nearly 40 GW of hydropower capacity could be potentially deployed in the short to medium term in order to meet such an increasing power demand [69]. Individual SAPP countries are already strongly dependent on hydropower production: Zambia and Mozambique rely on hydro for 80% of their electricity generation, also exporting their hydroelectric energy, while Zimbabwe hydro-power accounts for up to 60% of the total production (based on IEA data https://www.iea.org/). Due to low operational expenditures and high load factors, hydropower is identified as a cost-effective way to rapidly increase renewable energy uptake [74] and to offer an environmentally less harmful alternative to the fossil fuel electricity generated in the SAPP.

The *Zambezi River Basin* (ZRB), within the fourth-largest basin in Africa in terms of surface [75], covers approximately 1.4 million km$^2$, has a total length of 2,574 km, and is the largest basin in Southern Africa extending across many SAPP countries. Periods of prolonged droughts hinder the production of electricity due to the scarcity of water in the reservoir, while on the other hand, extreme flooding events put at risk dam safety and elevate downstream flood risk. Consequently, hydropower operators and river basin managers face a chronic challenge of balancing trade-offs between maintaining high reservoir levels for maximum power production and ensuring adequate, or low enough, reservoir storage volume in order to avoid risks from incoming floods. Evaporation losses further reduce the water availability in the basin: besides the natural losses that account for about 20% of the precipitation [76], the evaporation from large hydropower reservoirs exceeds 10% of the mean annual river flow. These water losses increase the risk of shortfalls in power generation and significantly impact both in-reservoir and downstream ecosystem functions. In the future expansion plans of the SAPP [77], hydropower development of the ZRB plays a crucial role, four new dams are planned to come into operation before 2026, representing a total installed capacity of 6.35 GW, aiming at more than doubling the existing capacity of 4.91 GW [78]. Run-off river plants exist in the considered study area, but their installed capacity is negligible with respect to the considered reservoir system, and for this reason left out of the scope of the modelling effort carried out in this work. The ZRB and the SAPP were selected as a case study because they represent an extremely suitable ground for testing our approach, different countries with different electricity generation mix, interconnected with each other and with a relevant river basin flowing through many of the involved countries, with dams distributed among them. It is an area widely studied by the scientific community in terms of the Water-Energy Nexus [79] and in addition to that, the participation of the authors in the DAFNE project (https://dafne.ethz.ch/) made data easily accessible making it the final decision for the case study.

### 3.1. Model definition, assumptions, and exogenous data

The definition and setup of the *Calliope_Hydro* model require the definition of the following elements.

**3.1.1. Purpose of the model.** Since the focus of the study consists in the improvement of hydropower modelling, the *Calliope_Hydro* model is set up to perform a system operation analysis, hence deriving optimal hourly power dispatch scenarios once electricity demand and

installed capacity for power generation and transmission technologies are known and exogenously imposed to the model. The model minimizes the overall operational cost of the energy system (mostly due to fixed and variable operation and maintenance costs, including fuel costs), determining the optimal power dispatch strategy in compliance with the technical constraints (natural resource availability, plants ramping constraints, etc.).

**3.1.2. Space and time scopes.** In the *Calliope_Hydro* model, only a selection of SAPP countries crossed by or linked to the Zambesi River Basin is included within the model's boundaries (Fig 4), each represented as one unique node, except for Mozambique, due to higher data availability compared to other countries. Angola, Tanzania and Malawi were excluded since they are not connected to the common grid, while Lesotho and Swaziland are excluded from the modelled network because of their negligible contribution to the energy generation, equal to the 0.32% of SAPP total capacity. Even if not completely outside the ZRB region, the Democratic Republic of Congo is also excluded from the *Calliope_Hydro* model because its total energy demand is mostly based on domestic power production, with imports and exports of the same order of magnitude of the statistical error (based on IEA data). As for the standard Calliope model, the time resolution of the *Calliope_Hydro* model is one hour, the hydrological variables are also represented with hourly timestep, and the inflow data, available with a monthly resolution, were equally distributed among the hours of each month.

**3.1.3. Supply side constraints.** Such constraints include technical and economic data related to the available power generation and transmission technologies, and the available resources (eventually geo-referenced, depending on the definition of nodes per country). Concerning the available power technologies, we model techno-economic parameters according to IRENA [54] and IEA [80] references. Technical and economic data for the available technologies in each region are reported in the S1 File, derived based on national energy companies reports [81–87]. The availability of renewable energy sources in all the modelled regions is derived as hourly regional averaged capacity factors from the open-source dataset www.renewables.ninja [88]. The South African Power Pool map available in the SADC Report [73] is assumed as the reference for defining the transmissions network among the modelled regions, as reported in Fig 4. The distribution network within each country is not modelled,

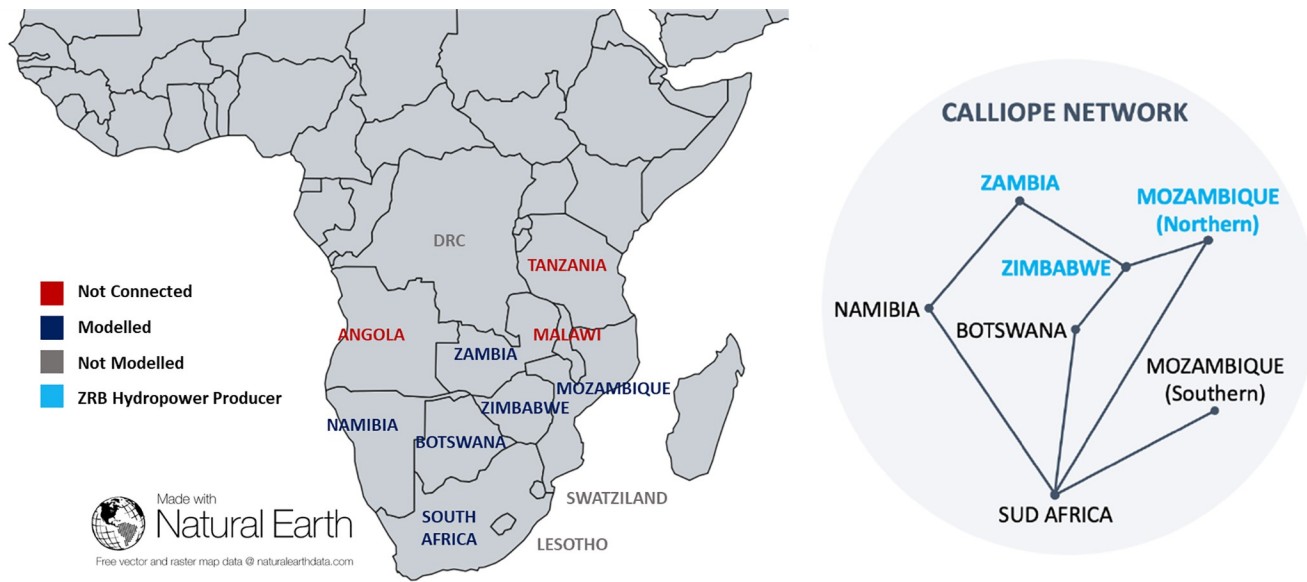

**Fig 4. Southern Africa Power Pool (SAPP) countries and modelled network.** Countries excluded from the analysis are highlighted in red, while countries part of the Zambesi River Basin (ZRB) are highlighted in light blue. Made with Natural Earth.

since each region is modelled as a single node (with the only exception of Mozambique, divided into northern and southern regions).

**3.1.4. Demand side constraints.**   Hourly timeseries of electricity demand are derived with different techniques depending on the available data of recorder electricity consumption: yearly aggregates are usually available, while hourly power demand may be difficult to be found, especially for developing countries. For Mozambique, Namibia, Zambia, and South Africa different load curves representative of different typical days of the year are available [89–92] (e.g. winter and summer days or weekday and weekend days). The shape of the available power profiles for the typical days are assumed as constant and scaled for the other days of the year based on the available data (e.g. weekly mean, monthly or seasonal electricity consumption). Where load curves are not available (Zimbabwe and Botswana), demand profiles are computed referring to neighboring countries and scaling properly based on national average electricity consumption [85, 93].

A further limitation of the modelling approach is the single node representation of the energy demand of each country, exception made for Mozambique, characterized by two demand nodes. In energy system modelling, the level of aggregation of country energy demand depends on the aim and scope of the analysis and, secondly, the availability of detailed energy demand data. Increasing the number of demand nodes characterization may have resulted in more detailed and accurate results. For the context of analysis, one demand node per country was considered a fair trade-off between the data paucity that characterizes the Sub-Saharan context and the results' accuracy. This choice is, anyway, in line with the most recent literature on the subject [94, 95].

**3.1.5. Multiple cascade water reservoirs.**   The Zambezi River Basin is modelled within the *Calliope_Hydro* model as a network of water rivers (Fig 5) linking the four main ZRB dams (and the related hydropower plants) as described in sub-section 2.1: *Itezhi-Tezhi* ("ITT", 120 MW), *Kafue Gorge* ("KGU", 990 MW), *Kariba* ("KA", 1.8 GW) and *Cahora Bassa* ("CB", 2 GW). The four dams are considered "large dams" according to the World Commission of Dams [96], being higher than 15 m. **Table 1** reports details on the four dams. Notably, the dams are located in different countries and may be linked with each other. Inflow data (the SW block in Fig 5) from 1986 to 2005 are extracted from the ADAPT project [97] using the following gauging stations: Kafue Hook Bridge, Victoria Falls IN, Great East Road Bridge, and Mangochi. Data about reservoirs' operational constraints as maximum and minimum release are retrieved from Gandolfi et al. [98], the *Zambezi River Authority* [78], and *Hidroeléctrica de Cahora Bassa* [99]. The storage initial values of the simulation period are extracted from dams' operational rule curves [100]. In order to compute evaporation losses and variable hydraulic head in the external loop evaporation rates are provided by Beilfuss and dos Santos [101] while reservoirs' level-storage and surface-storage curves are derived from World Bank [102]. For what concerns environmental and minimum flow regulations, only ITT has minimum flow regulations, but they were not implemented as out of the scope of this work.

**3.1.6. Hydrological parameters.**   To capture the full hydrological dynamic of ZRB's reservoirs and to cover its possible hydrological variability, an overall time horizon of two years is defined, and different water inflow scenarios are explored. Specifically, twenty years of historical monthly water inflow data from 1986 to 2005 are considered, divided equally per each hour of the month, defining 10 scenarios of two-year periods. Besides the inflow patterns, scenarios are also defined and influenced by the reservoir storages levels at the initial timestep, derived based on historical storage inputs data.

## 3.2. Scenarios definition

Fig 6 shows cumulated inflow for each historical year of available data, from 1986 to 2005 for each of the four reservoirs, shown in different colors.

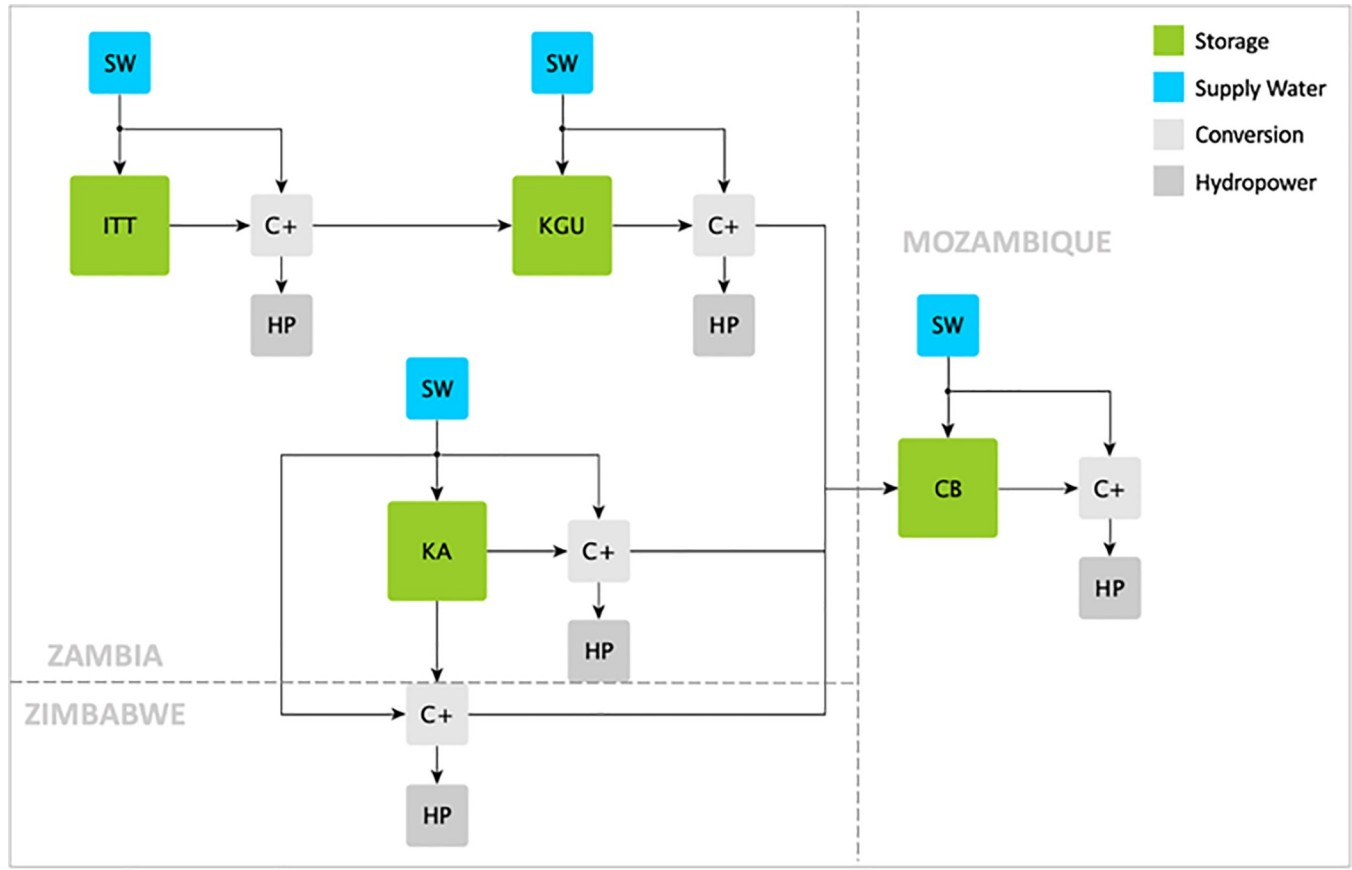

**Fig 5. Zambesi River Basin (ZRB) modelling scheme within the *Calliope_Hydro* model.**

In order to test the robustness of the model, different 2-years periods are considered to represent diverse hydrological regimes:

- **1993–1994:** this period presents a steep decrease in inflow volume from the first year to the second one.

- **1995–1996:** this period is selected as particularly dry.

- **1998–1999:** this period is identified as particularly wet.

- **2000–2001:** this period presents a steep increase in inflow volume from the first year to the second one.

**Table 1. Characteristics of the four considered dams.** Sources: [101, 103, 104].

| | Coordinates (Lat; Lon) | Capacity [MW] | Annual Inflow [km³] | Dam Height [m] | Live Storage [km³] | Dead Storage [km³] | # Turbines | Turbine Type | Turbine Efficiency |
|---|---|---|---|---|---|---|---|---|---|
| Ithezithezi | -15.763; 25.970 | 120 | 9.4 | 62 | 4.9 | 0.7 | 2 | Kaplan | 89% |
| Kafue Gorge | -15.808; 28.421 | 990 | 7.7 | 50 | 0.8 | 0.2 | 6 | Kaplan | 89% |
| Kariba | -16.522; 28.759 | 1830 | 40.2 | 97 | 64.7 | 116 | 12 | Kaplan | 89% |
| Cahora Bassa | -15.586; 32.704 | 2075 | 78.7 | 163 | 51.7 | 12.5 | 8 | Kaplan | 89% |

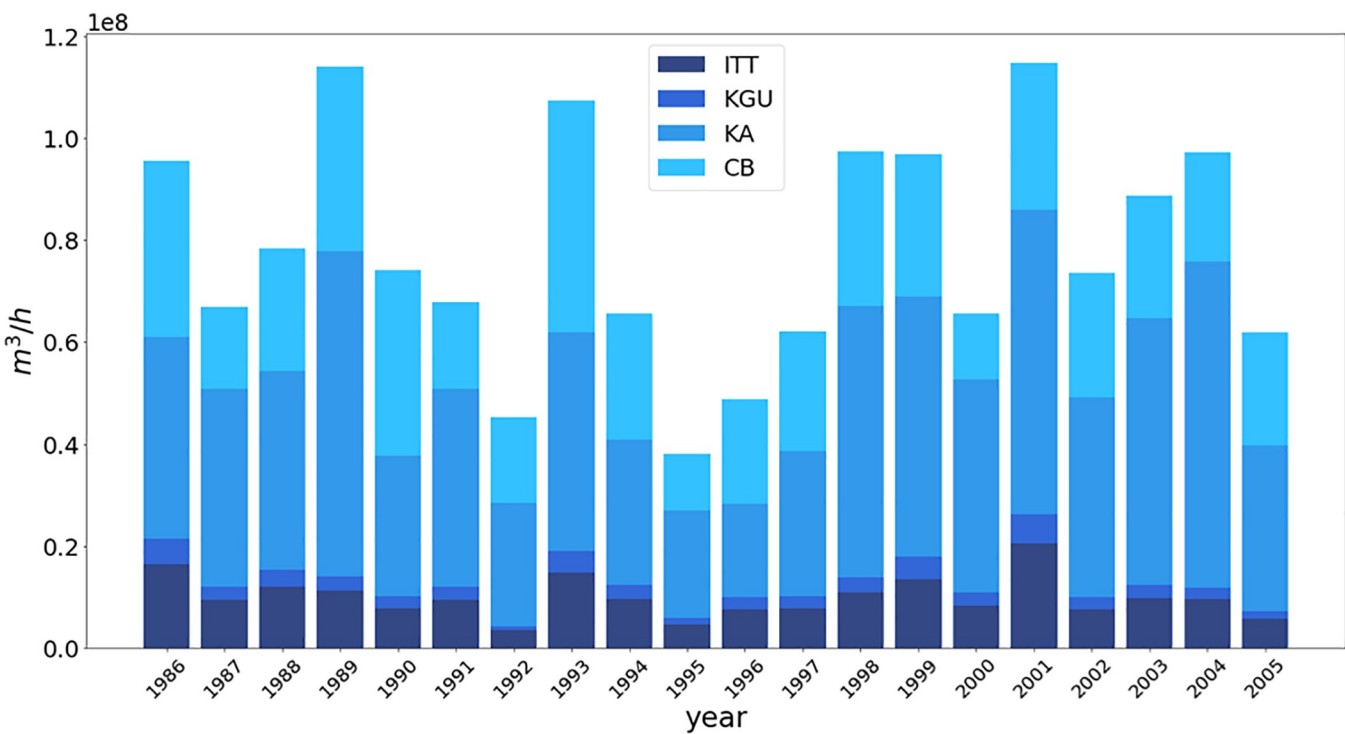

**Fig 6. Cumulated historical inflow from 1986 to 2005 for Zambesi River Basin reservoirs.**

Results obtained with the *Calliope_Hydro* model are then compared with the ones obtained from the *Calliope_Base* model, where hydropower is modelled without the advances proposed in this paper. The countries' load demand among the scenarios is not changed. To avoid unrealistic operation patterns of Coal Power Plants, two constraints are added: (1) a minimum technical power output and (2) ramping time constraints following IRENA's work on thermal power plant flexibility [105].

In accordance with the most recent international policy direction, the entire SAPP is modelled as a single, perfectly informed energy regulatory authority, which minimizes the overall operational costs of energy generation for the entire area as a single market, rather than optimizing the involved countries singularly.

## 4. Results and discussion

The overall amount of electricity produced from hydropower in the six SAPP countries and the two-years periods estimated by both *Calliope_Base* and *Calliope_Hydro* is larger (+104% and +84%) with respect to the IEA statistical observation. This discrepancy is due to the nature of the Calliope framework, which simulates an ideal power system ruled by a perfectly informed regulator, hence defining the least-cost electricity dispatch strategy with perfect foresight, and assuming a fully anelastic energy demand. Although these assumptions are very common in energy modelling practice [47], they still represent a limitation to the study, even though this limitation does not compromise the final aim of this work, which is adding constraints related to the physical nature of the reservoir hydropower to an energy system model. In the current context, the SAPP is indeed managed by different regulators with poor communications among them, resulting in management of water bodies and thermal power plants different with respect to the ideal estimation of the energy model [106]. Notwithstanding this, it emerges how the use of *Calliope_Hydro* allows a reduction of the error of the model of 20%.

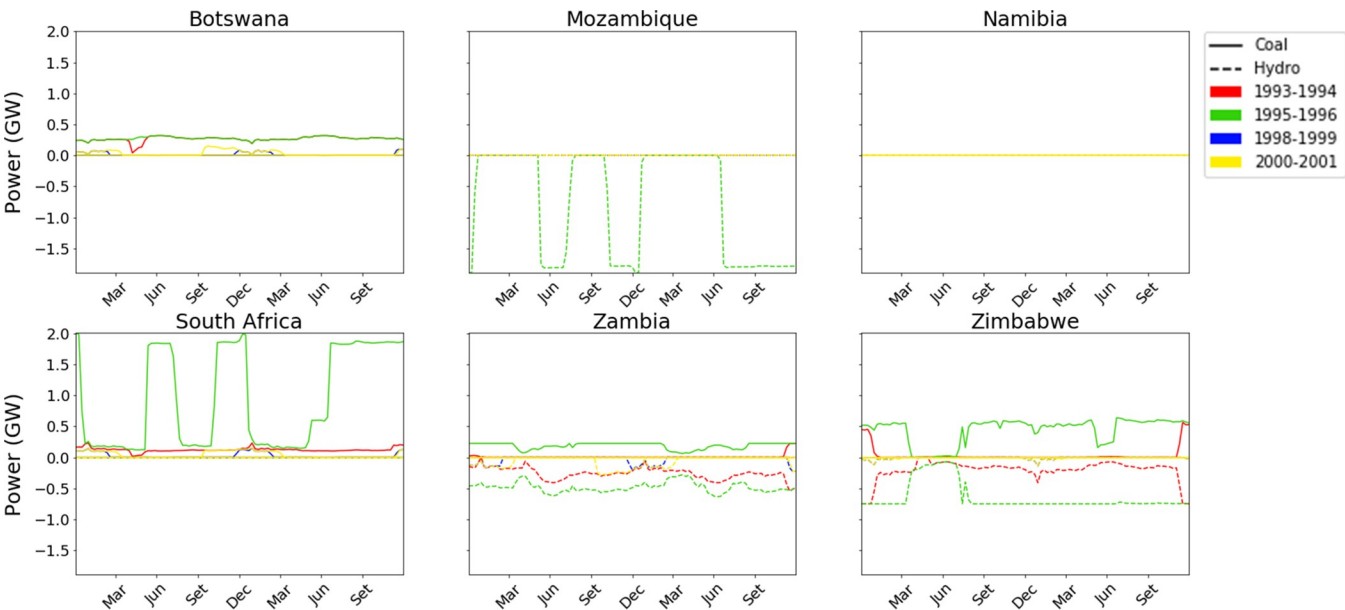

**Fig 7. Differential dispatch of the six countries of the power pool, as the difference in power produced between *Calliope_Hydro* (in the four scenarios) and *Calliope_Base*.**

Comparing the outputs of the two Calliope models, three different trends emerge: i) Botswana has no hydropower installed capacity, resulting in zero hydropower production, differences are present anyway in their electricity import strategy, that is dependent on the availability in neighboring countries, a different import availability hence influences the domestic coal power production; ii) the overall hydropower production of Namibia and South Africa derived by the two models is the same: this is due to the fact that those countries have no hydropower plants within the ZRB modelling scope, and have been hence modelled with *Calliope_Base* approach, again differences are present due to different import/export strategies and consequently the domestic production; finally iii) Mozambique, Zambia and Zimbabwe results of the two Calliope models present differences in the estimated power production over the two years periods. Such differences are amplified in the dry scenario "95–96", where the *Calliope_Hydro* model better captures the water scarcity of the analyzed years, showing that using a dynamic allocation of resources that depend on availability is more representative of the physical behavior of the hydropower plants. In fact, if only Mozambique, Zambia, and Zimbabwe are considered the reduction of the error in overestimating hydropower production reaches a value of 25% (from +107% to +82%), and if considering only the driest scenario, the overestimation of hydropower production goes from +119% to +18%.

Moving from the overall produced electricity to a dispatch analysis, we illustrate in Fig 7 the difference of hourly power production by technology between the two Calliope models for each modelled country in the four analyzed periods.

In the figure, positive values indicate a surplus of electricity produced by a certain technology derived by the *Calliope_Hydro* model with respect to the *Calliope_Base* model (and the opposite holds for negative values of the y-axis). The *Calliope_Hydro* model reveals a generally lower availability of hydropower compared to the *Calliope_Base* version, especially in scenarios 93–94 (8.6%) and 95–96 (35.2%), confirming the high sensitivity of hydropower production with respect to water availability in dry periods. The models' simulations suggest that the two pivotal technologies of the SAPP are coal-fired power plants and hydropower, as confirmed by

the International Energy Agency Data [107]. Indeed, even if nuclear technology (located in South Africa) is the third main source of electricity, it is not influenced by the different modelling approaches to hydropower, being suited for "baseload" activity due to techno-economic reasons.

To better grasp the effect on the electricity system of each country, in Fig 8 we propose the same analysis of Fig 7, but reporting the hourly differential power dispatch divided by the electricity demand in the same hour.

By a closer look at the hourly dispatch strategies in each region determined by the two models, significant discrepancies emerge between the two modeling approaches. Zimbabwe and Zambia present reductions of hydropower production in all four scenarios, while Mozambique shows a reduction only in the dry scenario (Fig 9). This suggests that the introduction of water mass balance influences the reservoir management: indeed, Zambia and Zimbabwe regions manage their basins (ITT, KGU, and KA) to guarantee adequate inflow of water to Mozambique's CB dam, This phenomenon is due to the assumption of full cooperation between the states, as observed in [106]. The other countries are not affected by hydropower reduction because their basins were not modelled with the approach here proposed.

Similarly to the results in Fig 7, the driest scenario 1995–1996 is the one presenting the largest differences between the two modeling approaches. It has been highlighted how the next century may incur in the rise of average atmospheric temperatures and decrease of precipitations [108], representing a serious challenge for countries heavily relying on hydropower for their electricity supply [109], and posing a risk that has to be taken into account when dealing with energy planning. Because of these reasons, scenario 1995–1996 is assumed as the reference for a deeper analysis. The analysis of the power dispatch strategy of the six countries over this period as modelled by *Calliope_Base* and by *Calliope_Hydro* (Fig 9) allows the comparison of the hourly power production yields by technology in each region and the electricity trades among them, in the form of imports and exports.

Our results show an overall increase in production from coal power technology, which balances the reduction of hydropower production. This trend is observed in the countries where

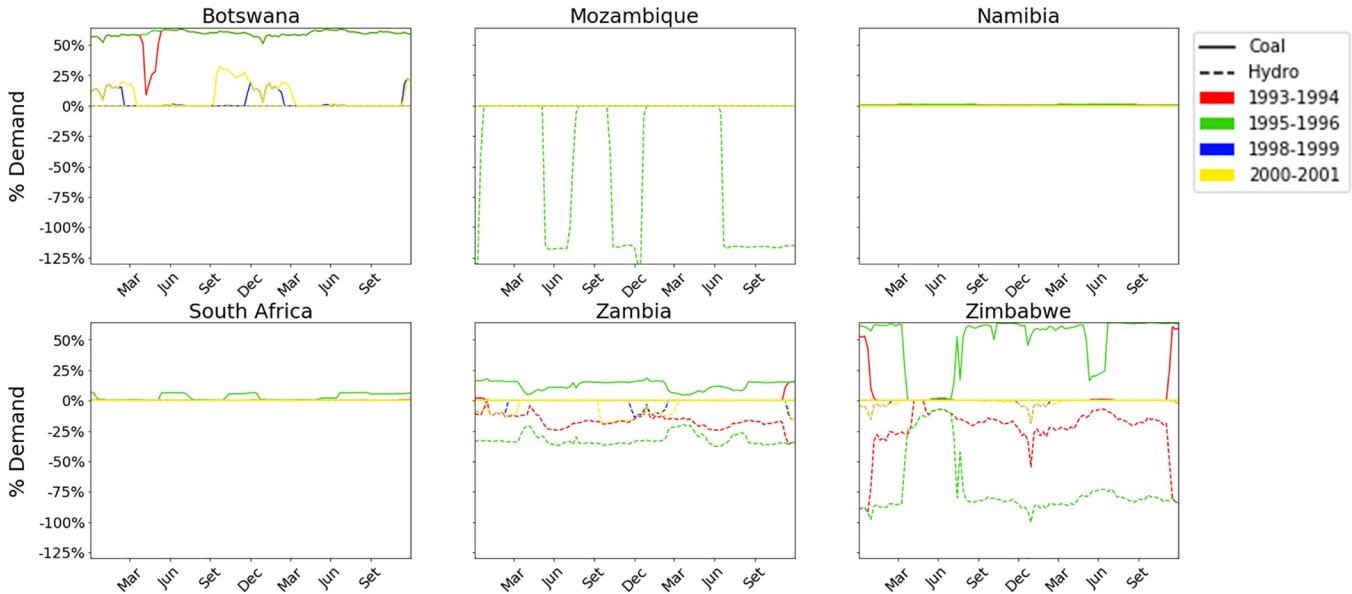

**Fig 8. Differential dispatch of the six countries of the power pool, as the difference in power produced between *Calliope_Hydro* (in the four scenarios) and *Calliope_Base*, relative to the national electricity demand.**

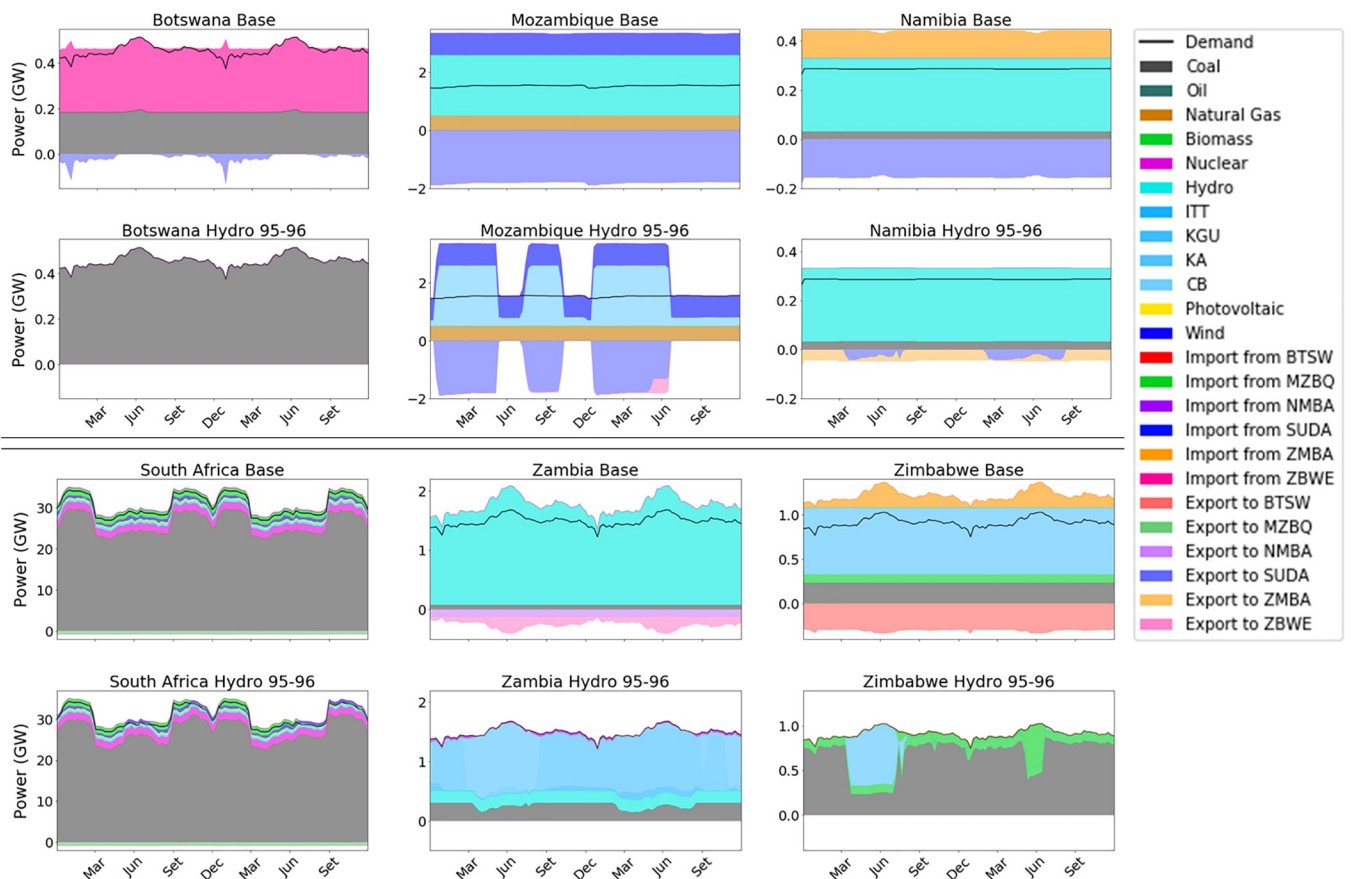

**Fig 9. Dispatch analysis of *Calliope_Base* and *Calliope_Hydro* over the period 1995–1996 (driest scenario).**

hydropower is not present or less abundant, namely Botswana and South Africa, because the first impact of reducing hydropower production in hydro-producing countries (Mozambique, Zambia, Zimbabwe) is a reduction of the energy available for export, resulting in the need to increase domestic production of non-hydro-producing countries with available resource and technologies able to modulate their loads. Extreme droughts further reduce hydropower productions, inducing also hydro-producing countries to increase their power production from coal (see the results for Zambia and Zimbabwe in 9). Coherently, an overall decrease in export from hydropower-producing countries is observed. For a broader graphical analysis of the different outputs of the models and their comparison, the reader can refer to the S1 File of this work.

The reader can notice that the results in Fig 9 present some abrupt changes in hydropower production, specifically in subplots *Mozambique Hydro 95–96* and *Zimbabwe Hydro 95–96*. Such abrupt changes are due to the nature of the energy model's optimization nature, the objective function is the minimization of the Net Present Cost of the entire system, and in seeking this objective it makes use of the dams' basins with this scope, putting economical benefits in front of dam's operating rules. Indeed, better characterization of constraints of dam's operating rules would help enhance further the effectiveness of this approach, which nonetheless already shows improvements with respect to the basic approach to the matter.

It is worth noting that including in the analysis also non-hydropower uses of water, would shift even more in these results towards a lower availability of hydropower in the system, given

the fact that considering it would decrease the amount of water available in the system for power production purposes.

Another limitation of the proposed case study, even though common in similar studies [94, 95], is the assumption of a single market for the entire power pool. Each of the modelled countries, even though exchanging with the neighboring ones, is not yet part of a single SAPP energy market. This assumption partially influences the results in terms of exchanged power, and in turn, of produced power. The model optimizes on the minimization of the overall cost of operating the entire system, in this way the energy mix of a country is influenced by its possibility to fulfill the energy needs of another country. This results in higher use of technologies with lower operation costs, that are employed to cover the demand of the countries that have more costly technologies in their production park, inside the limits of transmission capacity. Nonetheless, this does not hinder the value of the proposed methodology, which is still demonstrated to be valid by the case study. To sum up, the most relevant benefit of the proposed approach is related to the possibility of more accurately determining the availability of the hydropower source in each plant by closing the water mass balance, including natural and anthropogenic contributions. This represents a significant step forward in characterizing the actual availability of hydropower sources compared to the traditional approach.

In order to replicate the proposed approach to other large cascade reservoir basins, both in Africa or worldwide, it would only be necessary to know the geography of the basin, meaning the interconnections between the dams, the characteristics of the considered dams, as listed in **Table 1** for this work, and the inflows expected for the modelling period.

## 5. Conclusions

Most widely adopted power nodes energy systems models supporting energy planning strategies represent hydropower reservoirs as dispatchable thermal plants, disregarding the complex physical nature of multi-cascade water reservoirs. This approach introduces relevant model biases as it fails in capturing the dynamics of water availability in hydropower reservoirs, which is driven by the natural hydrologic variability.

To close this gap, we advance the Calliope modeling framework by adopting a new modelling approach aiming to better reproduce the real dynamics of multi-cascade hydropower reservoirs. As a proof of concept, our new model, called *Calliope_Hydro*, is tested in the case study of the Zambezi River basin in the Southern African Power Pool, across a wide range of hydrologic conditions, from extremely wet to extremely dry periods.

Numerical results suggest that the proposed modelling approach is successful in better capturing the dynamics of hydropower reservoir cascades, in contrast to the "stand-alone" role of each cascade adopted in traditional energy systems models, even though still showing discrepancies from observed IEA data. The second improvement observed in our results is the high sensitivity of the model to different hydrologic periods. This is of utter importance in supporting ongoing energy systems planning due to the projected rise in temperatures and decrease in precipitations in southern Africa over the next decades.

The model presents nonetheless some limitations that need further improvements. The spillage phenomenon is still not represented in its entire complexity and leaves space for improvement in the proposed framework. The open-source and technology-detailed nature of Calliope nonetheless allows for potential solutions to this shortcoming by implementing new ad-hoc rules in the optimization framework that will make it possible to represent the spillway technology. Multi-Stakeholder Management is not yet encompassed in the proposed approach, this is due to the complex nature of assigning monetary value to other uses of water apart from power production. Natural water is often unvalued or undervalued: the real economic value of

water is hardly accurately quantifiable, as well as the costs and benefits deriving from its marginal usage. On the other hand, the same can be more easily done for energy.

The experiment presented in this work was carried out in the energy modelling framework Calliope, a power nodes model widely known and adopted in the open-energy-modelling community [47, 48, 62, 63, 65], but the issue of how to better represent reservoir hydropower is cross-cutting to the community and every linear optimization based power nodes model could benefit from this advancement implementing the same architecture in their framework. What is more, the proposed advancement, together with another previous work from the authors [71]—where heat pumps were modelled following a similar logic—suggests that the process of external iteration (Fig 3) could be a potential good practice to preserve linearity in energy system modelling.

## Supporting information

**S1 File. Modelling parameters and supplementary results.**
(DOCX)

**S1 Appendix.**
(DOCX)

## Acknowledgments

We would like to thank Martina Daddi and Alessandro Barbieri for their contribution to developing initial numerical experiments.

## Author Contributions

**Conceptualization:** Nicolò Stevanato, Matteo V. Rocco, Matteo Giuliani.

**Methodology:** Nicolò Stevanato.

**Project administration:** Nicolò Stevanato.

**Software:** Nicolò Stevanato.

**Supervision:** Andrea Castelletti, Emanuela Colombo.

**Visualization:** Nicolò Stevanato.

**Writing – original draft:** Nicolò Stevanato, Matteo V. Rocco, Matteo Giuliani, Andrea Castelletti, Emanuela Colombo.

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
