## [Decision Letter · Decision Letter 0]

10 Feb 2021

PONE-D-20-40043

Advancing the representation of Reservoir Hydropower in Energy Systems Modelling: the case of Zambesi River Basin

PLOS ONE

Dear Dr. Stevanato,

Thank you for submitting your manuscript to PLOS ONE. After careful consideration, we feel that it has merit but does not fully meet PLOS ONE’s publication criteria as it currently stands. Therefore, we invite you to submit a revised version of the manuscript that addresses the points raised during the review process.

Note that we expect a MAJOR revision. Currently, the novelty and usefulness of the study remain unclear and the scientific quality and integrity are sometimes questionable, although it is a great contribution to open science that the code will be shared. Please find more detailed comments below.

We look forward to receiving your revised manuscript.

Kind regards,

Laura Scherer

Academic Editor

PLOS ONE

2. We note that Figure 4 in your submission contains map images which may be copyrighted. All PLOS content is published under the Creative Commons Attribution License (CC BY 4.0), which means that the manuscript, images, and Supporting Information files will be freely available online, and any third party is permitted to access, download, copy, distribute, and use these materials in any way, even commercially, with proper attribution. For these reasons, we cannot publish previously copyrighted maps or satellite images created using proprietary data, such as Google software (Google Maps, Street View, and Earth). For more information, see our copyright guidelines: http://journals.plos.org/plosone/s/licenses-and-copyright.

(1) You may seek permission from the original copyright holder of Figure 4 to publish the content specifically under the CC BY 4.0 license. 

3. Please ensure that you refer to Figure 3 in your text as, if accepted, production will need this reference to link the reader to the figure.

4. Please upload a copy of Figure 9, to which you refer in your text on page 15. If the figure is no longer to be included as part of the submission please remove all reference to it within the text.

5. Thank you for stating in your Funding Statement:

"MG and AC were partially supported by DAFNE-Decision Analytic Framework to explore the water-energy-food Nexus in complex transboundary water resource systems of fast developing countries research project funded by the Horizon 2020 programme WATER 2015 of the European Union, GA 690268. Data from the mentioned project were used."

Additional Editor Comments (if provided):

Be careful in your wording as to not claim novelty of previous contributions and not falsely suggest the solution of issues that remain unresolved (like the mismatch in the temporal resolution). Extend the literature review to cover more models that already fully integrate energy and water models, and clarify the contribution of your study. Justify also the choice of the Calliope model as a basis.

Critically discuss the limitations of hard-linked energy and water models in general and your approach specifically. In addition, acknowledge the high bias in your model, go more in depth with a validation against observations such as from the IEA, and compare the bias in your model to that of other fully integrated energy and water models.

Clarify the scope (e.g. only storage hydropower or also run-of-river hydropower) and the  methodology (e.g. the reservoir operating rules and the consideration of non-hydropower water users of multi-purpose reservoirs).

Indicate the sources of all input data to make the study more reproducible.

Explain strange results like the abrupt changes for power production.

Let a native speaker carefully proofread the manuscript.

Reviewers' comments:

Reviewer's Responses to Questions

**Comments to the Author**

1. Is the manuscript technically sound, and do the data support the conclusions?

Reviewer #1: No

Reviewer #2: Yes

2. Has the statistical analysis been performed appropriately and rigorously? 

Reviewer #1: Yes

Reviewer #2: Yes

3. Have the authors made all data underlying the findings in their manuscript fully available?

Reviewer #1: Yes

Reviewer #2: No

4. Is the manuscript presented in an intelligible fashion and written in standard English?

Reviewer #1: No

Reviewer #2: Yes

5. Review Comments to the Author

Reviewer #1: Summary:

This paper presents a framework for considering hydropower generation in power system models. The framework is demonstrated on the transboundary Zambesi River Basin and the South African Power Pool to show the proposed framework's benefits.

Overall, the topic of water-energy system modeling is interesting and of high relevance to many regions worldwide. However, it is not clear what the contribution of this study is compared to previous studies that consider hydropower and river system variability in modeling power systems. This study uses "full water-energy integration," which is not new to this area of research. Furthermore, the case study application is not well described, especially from the river system part. Moreover, this work includes significant assumptions and simplifications on the power system model, resulting in serious questions on the whole modeling exercise. I hope you find them useful in improving the modeling framework and the manuscript. This work can be published, but only after SIGNIFICANT modifications to the manuscript and case study application. Below find my detailed comments.

Comments:

• Lines 11-12: This opening statement is problematic. Several previous studies already connected power system models to water resource system models using a variety of approaches. In the literature review section, the authors mention many examples of such applications. You cannot simply ignore all these efforts by saying, "represented as any other thermal power plant." The two papers below are further examples of integrated water-energy modeling.

Gonzalez, J. M., Tomlinson, J. E., Harou, J. J., Martínez Ceseña, E. A., Panteli, M., Bottacin-Busolin, A., … Ya, A. Z. (2020). Spatial and sectoral benefit distribution in water-energy system design. Applied Energy, 269(May), 114794. https://doi.org/10.1016/j.apenergy.2020.114794

Sterl, S., Vanderkelen, I., Chawanda, C. J., Russo, D., Brecha, R. J., van Griensven, A., … Thiery, W. (2020). Smart renewable electricity portfolios in West Africa. Nature Sustainability, 3(9), 710–719. https://doi.org/10.1038/s41893-020-0539-0

• Line 15: Having read the full paper, I still do not understand what novelty is shown in this paper. The authors use a full integration water-energy framework to simulate a power system. Neither considering hydropower in water-energy system modelling is new or using a full integration framework is. The analysis presented in this paper is interesting and can be considered for publication, but the authors need to avoid overreaching or claiming previously published contributions. See my comments below for further details.

• Line 24: Why are African countries singled out here? Water-energy modeling is relevant anywhere, not only in Africa.

• Lines 46-47: This is incorrect. Please acknowledge that some previous studies considered hydropower and hydrologic uncertainty in power system modeling. I provided two examples in an earlier comment, but the authors also have examples in the literature review section.

• It seems this paper is framed around a false assumption that no previous study considered hydrologic variability and reservoir storage and operation in simulating hydropower in power system modeling. This is a fundamental issue that needs to be addressed throughout the paper. This is not a novelty of this work.

• Lines 49-51: While this is true, the authors need to cite more (if not all) widely used water resource system modeling tools to strengthen this statement (e.g., MIKE, RIBASIM, RiverWare, WEAP, Pywr, etc.).

• Line 52: The authors mention temporal resolution as a limitation of previous studies. This is understandable. How did you improve on that? Later in the manuscript, I saw that your water model has a monthly time step while your power system model has an hourly time step. It does not seem that you tackled this issue of different temporal resolution at all rather than assuming uniform hourly river inflows throughout each month. This sentence gives a false indication that your work solves the issue of varying temporal resolutions. Please rewrite this sentence to clarify this.

• The structure of Section 1.1 is confusing. The section starts with some claims on water-energy system modeling limitations. Them the same section later presents examples for water-energy system models that include hydropower. Then some previous studies are presented in lines 89-107. Then a new set of studies is presented afterward. This section needs to be rewritten. It is not possible to identify the contribution of this paper based on this literature review section.

• Lines 74-75: This is simply incorrect. Please see my earlier comments.

• Line 76: Please define all abbreviations at their first occurrence.

• Lines 90-91: This is true for optimization-driven river system models. Rule-based river system models use system operating rules to drive reservoir storage and releases. The authors need to review both kinds of river system models.

• Lines 93-94: Many countries in Africa use hydropower for baseload. In that case, maximizing hydropower generation is perfectly okay. Please elaborate on where this approach is suitable and where it is not, based on the role of hydropower and whether hydropower dams are single-purpose or multi-purpose.

• Line 101-102: see my earlier comment on previous studies and that hydropower has already been represented considering hydrologic uncertainty.

• Line 104: I see that some of the terms used in Table 1 are defined in the next paragraph (e.g., soft and full integration). You need to define these terms earlier. Alternatively, you could add a note to the table to define these terms.

• Lines 123-124: Please elaborate on the implications of this shortcoming of soft linking for computational accuracy on both the water and energy system sides.

• Lines 139-141: This does not make any sense. Modeling a water-energy system requires knowledge of the interactions within a system. It has nothing to do with using a soft linking approach.

• Lines 162-166: The authors do not mention an important shortcoming of hard-linked water and energy models. Costs and benefits can be easily quantified for an energy system, but this is not the case for water systems. Natural water is often unvalued or undervalued; thus, the real economic value of water cannot be accurately simulated. Worldwide, cases of established water markets can hardly be found.

• It is not clear to me what the contribution of this paper is compared to previous studies that hard-linked water and energy models. I do not see any methodological improvement here. For example, Payet-Burin et al. use Mike 11, which is, to my knowledge, a water resource system model that considers the same constraints that the authors listed as a novelty of their study. This is just an example of many previous studies. See these other two studies below as well.

Gonzalez, J. M., Tomlinson, J. E., Harou, J. J., Martínez Ceseña, E. A., Panteli, M., Bottacin-Busolin, A., … Ya, A. Z. (2020). Spatial and sectoral benefit distribution in water-energy system design. Applied Energy, 269(May), 114794. https://doi.org/10.1016/j.apenergy.2020.114794

Sterl, S., Vanderkelen, I., Chawanda, C. J., Russo, D., Brecha, R. J., van Griensven, A., … Thiery, W. (2020). Smart renewable electricity portfolios in West Africa. Nature Sustainability, 3(9), 710–719. https://doi.org/10.1038/s41893-020-0539-0

• Line 202: you mention "water release constraints." Do these constraints include reservoir operating rules? For the ZRB, what are these rules? This is important for interpreting the results.

• Line 239: How do you calculate the hydraulic head? Do you consider the variability in tailwater elevation? Please clarify in the text.

• Figure 3 is not referenced anywhere in the paper.

• Line 300: You need to be careful when describing hydropower as sustainable. Large dams are known to have significant environmental impacts. Also, greenhouse gas emissions from the reservoirs of storage dams are significant, especially in the tropics. Hydropower might be environmentally better than conventional energy generation, but I would not go as far as "sustainable."

• Line 302: Largest in terms of what?

• Lines 304-305: I do not understand what this sentence means. Please revise the grammar.

• Section 3.1: What is the temporal resolution of your model? Do you have the same resolution for both the water and energy models? Please add this information to the sub-section "space and time scopes."

• Line 327: the authors mention "one unique node." I do not understand how this works. How did you consider the spatial constraint of the power system network if each country is represented as one node?

• Lines 344-345: This is a major issue of this modeling exercise and puts questions on its usefulness. Obviously, you cannot assess the added accuracy of your framework based on a lumped power system model that represents each country as a single node.

• Lines 353-354: This contradicts the first part of the sentence. The authors say there are recorded data for hourly electricity demand. Why are the yearly data not available?

• Lines 367-368: The authors mention "initial and final storage." Initial and final with respect to what? Are you referring to the whole simulation period? Please clarify.

• Line 368: The authors mention "dams operational curves." Are these dams single-purpose or multi-purpose? How do you simulate multi-purpose dams? For example, how is irrigation water supply represented in your model?

• Line 378: you mention "two-year periods." Why only two years? Why not simulate the entire 20 years? A two-year period is too short, especially in a system with multi-year storage reservoirs.

• Line 380: Please elaborate on how the initial and final reservoir levels are derived. Historical reservoir water levels should be obtained from data records and not based on operating rules.

• Line 384: How did you objectively select these periods to simulate as scenarios? You should base your selection on hydrologic metrics.

• Line 399: Your simulation scenarios involve periods that are more than 20 years old. How realistic is it to assume a single market for such an old period?

• Lines 407-408: These assumptions are inadequate for any study. A realistic simulation of water-energy systems is the ultimate goal. If your model cannot do it, then please just admit it as a limitation.

• Line 416-418: How did you simulate hydropower plants outside ZRB? Are they simulated based on the very simplified approach that your paper criticizes? Please clarify and justify.

• Figure 3: There are some strange patterns in this figure with regards to hydropower production. For example, there are rapid increases and drops in hydropower. Please explain why these happen. You need to provide the reader with details on reservoir operating rules to be able to interpret the results.

• Lines 491-492: This is simply incorrect. Many studies represented hydropower considering water availability constraints. The authors mention several examples of such studies. Please remove such erroneous claims.

• Lines 500-505: Please mention that your model still shows a high bias compared to observed IEA data. This raises the question of how significant your claimed accuracy gains are compared to this high bias.

• Line 512-514: This should be mentioned earlier in the methodology. It is still unclear how dam operating rules are implemented in your model and how non-hydropower water users are simulated.

• Lines 518-519: Again, this needs to be mentioned far earlier in the methodology. This is a major limitation. River systems are often not used for hydropower only.

• Generally, the article contains many language and grammar errors. I urge careful and perhaps professional proofreading.

Reviewer #2: In this paper the authors present an approach how to move from a Calliope_Base to a Calliope_Hydro modell to improve the representation of reservoir hydropower within open source energy systems modelling. In the beginning the authors presented a detailed overview of model background literature and they apply their new model to the Southern African Power Pool and compare it with the base model. Hence the research in this paper is very focused on one region/river basin. Nevertheless, the authors chose the title: “Advancing the representation of Reservoir Hydropower in Energy Systems Modelling”. However, I am missing this wider picture in the paper. They authors showed in a case-study that it is possible to move from Calliope_Base to Calliope_Hydro. But what’s the implication for the Energy Systems Modelling community? What is needed to apply you modell to other areas in Africa? Or globally? Furthermore, in my opinion the manuscript could benefit from some restructuring, especially the introduction (See detailed comments) and the method section to better highlight the differences between the Calliope_Base and Calliope_Hydro modell. In addition, the manuscript sometimes provides a lot of detail, while other parts are lacking detail (Literature review criteria/ discharge data). Therefore, I cannot accept the manuscript in its actual form for publication, but I am hoping that my comments can give a good starting point for a revision.

Abstract:

Line 16: Is the study now looking at hydropower generation in general or at “representation of storage hydropower” as mentioned in the highlights. Please clarify.

Line 23: The last sentence of the abstract is a bit misleading. Does this mean that the model is only applicable to Africa? Do only African reservoirs depend on water resources? In my opinion hydropower generates electricity but it cannot generate power.

Highlights:

Line 35: I would be careful with using the word “substantial” without any underlaying number

Introduction:

General comment:

In my opinion the introduction to the water-nexus problematic false very short. Then the authors present a literature review (without specifying the criteria), but for me as reader it is unclear what of the information is relevant for the aim of the study, as this is only defined after the Literature review. But the literature review itself provides a very detailed overview. Thanks. But here one could argue that not all this information is needed to understand the aim of the study. In addition, I have the following specific comments:

Line 37: You are citing SDG 6 and SDG 7, but they have nothing per se to do with economics and poverty.

Line 40: If you are stating that the Water-Energy nexus is increasingly recognized and studied, it would be nice to provide a newer reference than 2015. I think a lot of research has been done since that.

Line 42: Ref 5 Why are you referring to the Japanese translation?

Line 42: Of which nexus? In my opinion Lines 37-40 are extremely short for a Water nexus introduction. I would suggest that the authors are a bit more specific which interconnection are relevant for this study.

Line 43: Largely? Wouldn’t that implement that it also works without water? I think its “inter alia” effected by climate variability and allocations to other uses, as for example also turbine efficiency and head can play a role for hydropower generation. Here again it would be nice to be more specific. What climate variability is relevant for hydropower and what are other users?

Line 45: Please explain what you mean by “non-linear nature of hydropower”.

Line 59: Structure of the introduction: So far you have mentioned the research gap but not what the aim of the study is, but now I am presented with a chapter “Literature review”. Why has this been done, why is it important?

Line 71: You know mention “Reservoir hydropower”, before you talked about storage hydropower. Do these terms mean the same for you? I haven’t seen an explanation what you mean with storage hydropower yet.

Line 75. Here a reference for this statement should be added.

Line 88: Here you state that the energy model OsEMOSYS is enhanced. Why do you enhance this model?

Line 109: What do you mean by: “Literature is anyway rich of works”

Line 169: Based on what criteria have you selected/ reviewed the literature? Is this overview meant to be comprehensive? More information is need.

Line 174: Here you state that you are looking at “Calliope” but in line 88 you stated that you are enhancing the OsEMOSYS model. Please clarify.

Line 186: What do you mean by energy carrier?

Line 193: Here you state that “40 GW of hydropower could be potentially deployed in this region” . But in the abstract, you write from a potential of 20,000 megawatts (MW). What is now correct?

Methods:

Here the other show the used the Calliope model. However, in my opinion the authors fail to appropriately highlight what has been existing previously in the model and what their novel contribution is.

As I understand the other show in section 2.1 the Calliope_Base and then in Section 3.1 the Calliope_Hydro model. In the results you put the focus on the difference between Calliope_Base and by Calliope_Hydro. But the word Calliope_Hydro is for the first time mentioned in line 317. Maybe it would be better to combine he 2 chapters to highlight the differences?

Are run-off river power plants completely ignored by the model? Or are their now run-off river power plants in SAPP?

Line 204: I agree that factors are influencing the electricity production. But by how much do they influence the head? For me the head is the result of slope/high difference + water level in the reservoir. The factors you described are only looking at water level in the reservoir.

Line 204: what do you mean by hydropower technologies? I think your factors are only relevant for reservoir hydropower but not for run-off-river hydropower.

Line 243: “that flows thought» wrong word?

Case-Study:

In my opinion not only the amount of electricity potential, but also location and sizes of reservoirs should be presented. I am still lacking an explanation why the SAPP region has been selected. The authors then give a lot of explanation about the power pools. But wouldn’t be the type of hydropower and typical hydropower operations schemes, turbine/dam types more relevant for the reader?

Line 307: So there are no environmental regulations / minimum flow regulations that the hydropower operators have to balance as well?

Line 307: By “adequate” you mean low water level?

Line 309: What is a large reservoir? Until now no information about reservoir size was provided. Does the “the mean annual river flow” refer to the river section in which the reservoir is located?

Line 311: Only downstream? What about the biodiversity in the lake?

Line 313: Please be consistent with the digits: 6.345 GW VS 4.91 GW

Line 348: Reference to the map source is missing.

Line 375: From where do you receive your hydrological data?

Line 383: Would be nice to get a chart of the river discharge?

Results and discussion:

The results are presented in a nice and detailed way. However, I am the “discussion” part could be improved. For example: How would the results change if different time periods would have been chosen? What’s the difference in uncertanty between the Calliope Base and by Calliope_Hydro? In the abstract you highlighted that your support hydropower management and planning capacity in Africa. Please make the link to this statement. What is needed to apply you modell to other areas in Africa? Or globally?

Conclusions: You now showed in a case-study that its possible to move from Calliope_Base to Calliope_Hydro. But what’s the implication for the Energy Systems Modelling community?

6. PLOS authors have the option to publish the peer review history of their article (what does this mean?). If published, this will include your full peer review and any attached files.

Reviewer #1: No

Reviewer #2: No

---

## [Author Response · Author response to Decision Letter 0]

21 Mar 2021

EDITOR COMMENTS:

1. Be careful in your wording as to not claim novelty of previous contributions and not falsely suggest the solution of issues that remain unresolved (like the mismatch in the temporal resolution).

This issue has been also raised from the Reviewers: we have provided in this rebuttal detailed responses and tackled such comments throughout the entire manuscript. Thanks for pointing this out. 

2. Extend the literature review to cover more models that already fully integrate energy and water models, and clarify the contribution of your study.

Thanks for the suggestion, chapter 1 has been enlarged and restructured. In particular, thanks to the Editor’s and Reviewer’s comments, we were able to understand that we had improperly stated our contribution in the manuscript. The contribution we propose is the enhancement of an energy system model alone: no integration between energy and hydro models is performed, thus framing our contribution in a different context. 

3. Justify also the choice of the Calliope model as a basis.

Thanks for highlighting this shortcoming, this aspect has been deepened in the discussion. Namely lines 199-203:

“Among the variety of existing open energy modelling frameworks [33], Calliope [30] is selected due to its bottom-up technology driven approach and full customizability of the involved energy conversion technologies [55], that makes it an ideal ground for testing the proposed approach. Furthermore, the framework is widely adopted for the study of national power systems [56–58] and interconnected power pools [59–62] in literature.”

4. Critically discuss the limitations of hard-linked energy and water models in general and your approach specifically. In addition, acknowledge the high bias in your model, go more in depth with a validation against observations such as from the IEA, and compare the bias in your model to that of other fully integrated energy and water models.

Thanks, the results section has been revised and the methodology section critically analyses the shortcomings of the approach.

5. Clarify the scope (e.g. only storage hydropower or also run-of-river hydropower) and the methodology (e.g. the reservoir operating rules and the consideration of non-hydropower water users of multi-purpose reservoirs).

The scope of the paper has been better clarified throughout the entire manuscript and the methodology explained more in detail. Thank you.

6. Indicate the sources of all input data to make the study more reproducible.

Please notice that in the original submission we already provided the references to all the input data assumed for setting up the model. However, in this revision we made more explicit reference to every data source. Thank you. 

7. Explain strange results like the abrupt changes for power production.

This has been taken care of, as also suggested by Reviewer #1. Specifically, a new paragraph has been inserted to explain such behaviours and give the readers all the necessary information.

Lines 549-555: “The reader can notice that results in Fig 9 presents some abrupt changes in hydropower production, specifically in subplots Mozambique Hydro 95-96 and Zimbabwe Hydro 95-96. Such abrupt changes are due to the nature of the energy model’s optimization nature, the objective function is the minimization of the Net Present Cost of the entire system, and in seeking this objective it makes use of the dams’ basins with this scope, putting economical benefits in front of dam’s operating rules. Indeed, better characterization of constraints of dam’s operating rules would help enhance further the effectiveness of this approach, that nonetheless already shows improvements with respect to the basic approach to the matter.”

8. Let a native speaker carefully proofread the manuscript.

Thanks for highlighting this. The manuscript has been carefully proofread before re-submission.

REVIEWER #1 COMMENTS:

1. Lines 11-12: This opening statement is problematic. Several previous studies already connected power system models to water resource system models using a variety of approaches. In the literature review section, the authors mention many examples of such applications. You cannot simply ignore all these efforts by saying, "represented as any other thermal power plant." The two papers below are further examples of integrated water-energy modeling.

• Gonzalez, J. M., Tomlinson, J. E., Harou, J. J., Martínez Ceseña, E. A., Panteli, M., Bottacin-Busolin, A., … Ya, A. Z. (2020). Spatial and sectoral benefit distribution in water-energy system design. Applied Energy, 269(May), 114794. https://doi.org/10.1016/j.apenergy.2020.114794

• Sterl, S., Vanderkelen, I., Chawanda, C. J., Russo, D., Brecha, R. J., van Griensven, A., … Thiery, W. (2020). Smart renewable electricity portfolios in West Africa. Nature Sustainability, 3(9), 710–719. https://doi.org/10.1038/s41893-020-0539-0

Thanks for pointing this out, we would like to take advantage of this first comment to point out how we have reformulated the contribution of our work in this revised manuscript. 

In the original submission we had improperly stated that the contribution of the work is a full integration of an energy model with a hydro model to better represent reservoir hydropower operation. Thanks to the Reviewer’s and the Editor’s comments, we understood how we had misled the reader into thinking that. The focus of the paper was instead to provide an enhancement of the representation of reservoir hydropower in an existing, widely adopted, energy modeling framework. 

The most relevant novelty introduced by our research is that we proposed a general approach that could be in principle reproduced in any other energy modelling framework based on power nodes formulation. On the other hand, the suggested literature references provide ad-hoc models and approaches for dealing with hydropower modelling without improving one existing and widely adopted energy modeling framework.

The paper opening statement refers to state of-the-art energy models, which are the target of our work. 

Finally, the suggested literature was nonetheless precious for enhancing the literature review and has been added in a new category of relevant literature for the work. The entire wording throughout the manuscript has been updated accordingly. 

2. Line 15: Having read the full paper, I still do not understand what novelty is shown in this paper. The authors use a full integration water-energy framework to simulate a power system. Neither considering hydropower in water-energy system modelling is new or using a full integration framework is. The analysis presented in this paper is interesting and can be considered for publication, but the authors need to avoid overreaching or claiming previously published contributions. See my comments below for further details.

In line with the answer to the previous comment, thanks to the Reviewer’s comment we understood how we had placed our contribution in an incorrect framework. The entire manuscript was revised accordingly, and wording adjusted to identify the novelty of the work in the field of solely energy systems models.

3. Line 24: Why are African countries singled out here? Water-energy modeling is relevant anywhere, not only in Africa.

The sentence was misleading, it has been revised and corrected. 

Line 25-27: “These improvements are useful to support hydropower management and planning capacity expansion in countries richly endowed with water resource or that are already strongly relying on hydropower for electricity production.”

4. Lines 46-47: This is incorrect. Please acknowledge that some previous studies considered hydropower and hydrologic uncertainty in power system modeling. I provided two examples in an earlier comment, but the authors also have examples in the literature review section.

Thanks for highlighting this, the suggested literature was added (Lines 182-187) and the sentence revised. Studies concerning hydrologic uncertainty exist, and here we refer to energy system models.

Lines 56-57: “with few exceptions, energy system models fall short in considering the holistic influence of hydrometeorological variability on bulk power systems [14].”

5. It seems this paper is framed around a false assumption that no previous study considered hydrologic variability and reservoir storage and operation in simulating hydropower in power system modeling. This is a fundamental issue that needs to be addressed throughout the paper. This is not a novelty of this work.

We agree with the Reviewer, the entire manuscript was revised, the contribution we propose is inherent to energy system models, as highlighted in answer to comment #1.

6. Lines 49-51: While this is true, the authors need to cite more (if not all) widely used water resource system modeling tools to strengthen this statement (e.g., MIKE, RIBASIM, RiverWare, WEAP, Pywr, etc.).

We agree with the Reviewer, thanks for pointing out. The sentence was rewritten. 

Lines 59-61: “On the other hand, in many state-of-the-art hydrological models (e.g. MIKE [16], RIBASIM [17], RiverWare [18], WEAP [19], Pywr [20], SWAT [21] among the others) the energy system is not comprehensively included […].” 

7. Line 52: The authors mention temporal resolution as a limitation of previous studies. This is understandable. How did you improve on that? Later in the manuscript, I saw that your water model has a monthly time step while your power system model has an hourly time step. It does not seem that you tackled this issue of different temporal resolution at all rather than assuming uniform hourly river inflows throughout each month. This sentence gives a false indication that your work solves the issue of varying temporal resolutions. Please rewrite this sentence to clarify this.

In the Space and time scopes section (Lines 379-390), this issue has been better clarified, according also to other comments. The sentence has also been removed from the introduction to avoid giving wrong suggestions to the reader. 

Lines 387-390: “As for the standard Calliope model, time resolution of the Calliope_Hydro model is one hour, the hydrological variables are also represented in hourly timestep, and the inflow data, available with monthly resolution, were equally distributed among the hours of each month.”

8. The structure of Section 1.1 is confusing. The section starts with some claims on water-energy system modeling limitations. Them the same section later presents examples for water-energy system models that include hydropower. Then some previous studies are presented in lines 89-107. Then a new set of studies is presented afterward. This section needs to be rewritten. It is not possible to identify the contribution of this paper based on this literature review section.

Thanks for highlighting that this section was unclear. We had expanded Section 1.1 and better explained the role of the literature in the framework of the entire work.

9. Lines 74-75: This is simply incorrect. Please see my earlier comments

This sentence refers to a very specific category of models. The sentence has been restructured and a reference added.

Lines 85-88: “Typically, widely adopted only-energy models, both proprietary and open-source, deal with hydropower just by exogenously constraining its production based on the installed capacity and on a so-called capacity factor calibrated based on historical energy production data [35].”

10. Line 76: Please define all abbreviations at their first occurrence.

Thanks for pointing this out. The abbreviation has been defined.

Line 89: “[…] conducted by the International Renewable Energy Agency (IRENA) […]”

11. Lines 90-91: This is true for optimization-driven river system models. Rule-based river system models use system operating rules to drive reservoir storage and releases. The authors need to review both kinds of river system models.

Thanks, we changed the sentence into “is usually either set as an objective function to be maximized or implicitly included in the rule curve adopted to operate the reservoir without properly keeping into account the energy grid constraints”. Lines 103-105

12. Lines 93-94: Many countries in Africa use hydropower for baseload. In that case, maximizing hydropower generation is perfectly okay. Please elaborate on where this approach is suitable and where it is not, based on the role of hydropower and whether hydropower dams are single-purpose or multi-purpose.

We are not sure the comment is about line 93-94. If so, here we are revising some relevant literature not necessarily about Africa dams. In general, we do agree with the Reviewer that in most African pools, hydropower is contributing to baseload. This also applies to the SAPP considered in this paper. The advantage of a better characterization of the energy production/transfer component holds for any system including hydropower both single and multipurpose.

13. Line 101-102: see my earlier comment on previous studies and that hydropower has already been represented considering hydrologic uncertainty

Thanks for the comment, in the manuscript we do not refer to previous studies in general, but to existing power nodes energy system models, which are the scope of the present study.

14. Line 104: I see that some of the terms used in Table 1 are defined in the next paragraph (e.g., soft and full integration). You need to define these terms earlier. Alternatively, you could add a note to the table to define these terms.

We decided to postpone the reference to Table 1 after the definition of such terms. Thank you.

15. Lines 123-124: Please elaborate on the implications of this shortcoming of soft linking for computational accuracy on both the water and energy system sides.

Lines 123-124 were referred to the drawbacks of some currently adopted modelling approaches, which make use of non-linked energy and hydrological models. In those cases, the absence of a feedback dynamic between the two models, being it either soft- or hard- link, may return conflicting results. We have better expressed this concept in lines 135-136.

16. Lines 139-141: This does not make any sense. Modeling a water-energy system requires knowledge of the interactions within a system. It has nothing to do with using a soft linking approach.

We do agree with the Reviewer, the sentence was misleading, thanks for pointing out. We referred to the link of two different objective functions. Knowledge of the two systems is indeed fundamental.

Line 151-152: “However, it is not always easy to identify how to link the objective functions of the two different models.”

17. Lines 162-166: The authors do not mention an important shortcoming of hard-linked water and energy models. Costs and benefits can be easily quantified for an energy system, but this is not the case for water systems. Natural water is often unvalued or undervalued; thus, the real economic value of water cannot be accurately simulated. Worldwide, cases of established water markets can hardly be found.

 Thanks for pointing this out, we do agree with the Reviewer: this is a fundamental issue and has been inserted among the limitations of the hard linked models.

Lines 162-164: “Natural water is often unvalued or undervalued: the real economic value of water is hardly accurately quantifiable, as well as the costs and benefits deriving from its marginal usage. On the other hand, the same can be more easily done for energy.”

18. It is not clear to me what the contribution of this paper is compared to previous studies that hard-linked water and energy models. I do not see any methodological improvement here. For example, Payet-Burin et al. use Mike 11, which is, to my knowledge, a water resource system model that considers the same constraints that the authors listed as a novelty of their study. This is just an example of many previous studies. See these other two studies below as well.

• Gonzalez, J. M., Tomlinson, J. E., Harou, J. J., Martínez Ceseña, E. A., Panteli, M., Bottacin-Busolin, A., … Ya, A. Z. (2020). Spatial and sectoral benefit distribution in water-energy system design. Applied Energy, 269(May), 114794. https://doi.org/10.1016/j.apenergy.2020.114794

• Sterl, S., Vanderkelen, I., Chawanda, C. J., Russo, D., Brecha, R. J., van Griensven, A., … Thiery, W. (2020). Smart renewable electricity portfolios in West Africa. Nature Sustainability, 3(9), 710–719. https://doi.org/10.1038/s41893-020-0539-0

Thanks for reporting how the manuscript was unclear in highlighting the contribution. In line with the answers to comments 1 and 2, we have restructured the manuscript to differentiate among previous contributions by other authors, and this work contributions, and narrowed the scope of the proposed methodology, highlighting how it concerns only wide-spread already existing power nodes energy modelling frameworks. 

19. Line 202: you mention "water release constraints." Do these constraints include reservoir operating rules? For the ZRB, what are these rules? This is important for interpreting the results.

Thanks for the comment, the sentence needed a clarification, and has been added. Reservoir operating rules are not taken into account, we refer to maximum amount of water that can be processed by the turbines’ system.

Line 232-233: “[…] dam maximum and minimum operational level and related water release constraints (in terms of maximum amount of water processable) […]”

20. Line 239: How do you calculate the hydraulic head? Do you consider the variability in tailwater elevation? Please clarify in the text.

We changed the text to more accurately describe hydraulic head calculation. “The hydraulic head is computed as the difference between current water level in the dam and the elevation of the turbine axis. The tailwater effect is not considered.” Lines 273-274

21. Figure 3 is not referenced anywhere in the paper.

Thanks for pointing this out. It is now mentioned in Line 294

22. Line 300: You need to be careful when describing hydropower as sustainable. Large dams are known to have significant environmental impacts. Also, greenhouse gas emissions from the reservoirs of storage dams are significant, especially in the tropics. Hydropower might be environmentally better than conventional energy generation, but I would not go as far as "sustainable."

Thanks for the comment. We substituted "sustainable" with "environmentally less harmful". Line 354

23. Line 302: Largest in terms of what?

"in terms of surface" added in line 356, a reference has been added.

24. Lines 304-305: I do not understand what this sentence means. Please revise the grammar.

The sentence has been rewritten.

Line 358-360: “Periods of prolonged droughts hinder the production of electricity due the scarcity of water in the reservoir, while on the other hand, extreme flooding events put at risk dam safety and elevate downstream flood risk.”

25. Section 3.1: What is the temporal resolution of your model? Do you have the same resolution for both the water and energy models? Please add this information to the sub-section "space and time scopes."

Thanks for highlighting this shortcoming of the subsection, the missing information has been added.

Lines 388-391: “As for the standard Calliope model, time resolution of the Calliope_Hydro model is one hour, the hydrological variables are also represented in hourly timestep, and the inflow data, available with monthly resolution, were equally distributed among the hours of each month.”

26. Line 327: the authors mention "one unique node." I do not understand how this works. How did you consider the spatial constraint of the power system network if each country is represented as one node?

27. Lines 344-345: This is a major issue of this modeling exercise and puts questions on its usefulness. Obviously, you cannot assess the added accuracy of your framework based on a lumped power system model that represents each country as a single node.

We provide here the answers to comments 26 and 27.

We did this for two reasons: 

1. The increased spatial detail of the country energy system representation certainly improves the resolution and accuracy of the results derived by the energy model, however relying on a single node representation of national energy systems does not hinder the validity of the proposed improvement, related to the increased physical detail of the hydro reservoir systems. 

2. In power nodes models, the geographical detail achievable is constrained by the availability of demand data, since each node has to be characterized by supply technologies and demand to be met. Given the data paucity that characterizes Sub Saharan Africa's countries the best achievable solution was representing each country as a node of the model (except for Mozambique, for which data were slightly more abundant). Furthermore, the focus of the study was not on internal transmission but on the representation of reservoir hydropower in a multi node context, and the relative international power exchanges

28. Lines 353-354: This contradicts the first part of the sentence. The authors say there are recorded data for hourly electricity demand. Why are the yearly data not available?

Probably we explained ourselves not clearly enough, the sentence has been rewritten in order to be more understandable.

Lines 410-413: “Hourly timeseries of electricity demand are derived with different techniques depending on the available data of recorder electricity consumption: yearly aggregates are usually available, while hourly power demand may be difficult to be found, especially for developing countries.”

29. Lines 367-368: The authors mention "initial and final storage." Initial and final with respect to what? Are you referring to the whole simulation period? Please clarify

Thanks for highlighting this shortcoming, the missing information has been added.

Line 427: “The storage initial values of the simulation period are extracted”

30. Line 368: The authors mention "dams operational curves." Are these dams single-purpose or multi-purpose? How do you simulate multi-purpose dams? For example, how is irrigation water supply represented in your model?

We do not simulate multi-purpose dams, this is now mentioned at the end of methodology.

Line 334-338: “Multi Stakeholder Management, as mentioned in the introduction of this work, is another critical aspect of water resource management, and consists in enlarging the scope of the analysis to include other water-related sectors beyond power production needs. This would require the formulation of other water demands alongside the electricity one, to account for competing uses of water in the river basin.”

31. Line 378: you mention "two-year periods." Why only two years? Why not simulate the entire 20 years? A two-year period is too short, especially in a system with multi-year storage reservoirs.

The two years period is constrained by computational burden of the model, modelling a longer period would take an unjustified amount of time and the two years period was, in our opinion, enough to justify the validity of the proposed approach.

32. Line 380: Please elaborate on how the initial and final reservoir levels are derived. Historical reservoir water levels should be obtained from data records and not based on operating rules.

The sentence was misleading and has been corrected. The initial level of storage is derived from historical data. Then the storage is managed according throughout the modelling period as all the other operating variables, according to the optimization function, that in this case is the minimization of the NPC. 

Lines 439-440: “Beside the inflow patterns, scenarios are also defined and influenced by the reservoir storages levels at the initial timestep, derived based on historical storage inputs data.”

33. Line 384: How did you objectively select these periods to simulate as scenarios? You should base your selection on hydrologic metrics.

A figure (Fig 6) with the annual value of the inflow in the four scenarios has been included to better clarify the rationale behind our choice. Then In order to test the robustness of the model, four different 2-years periods are selected to represent: i) a steep decrease in inflow volume from the first year to the second one, ii) a particularly dry period, iii) a particularly wet period and iv) a steep increase in inflow volume from the first year to the second one.

34. Line 399: Your simulation scenarios involve periods that are more than 20 years old. How realistic is it to assume a single market for such an old period?

The rationale behind the four investigated scenarios is not to faithfully represent past behaviours, but test the behaviour of the approach in different hydrological conditions. This is also one of the reasons behind the discrepancy between IEA data and our results.

35. Lines 407-408: These assumptions are inadequate for any study. A realistic simulation of water-energy systems is the ultimate goal. If your model cannot do it, then please just admit it as a limitation.

 We agree with the Reviewer, this is indeed a limitation, which was already present in the Calliope_base framework, and that our approach partially contributed to reduce. We admitted the limitation in the manuscript, but we would like to highlight how our scope is not that of reproducing accurate scenarios of electricity operational dispatch, for which the single node representation is not suitable, but improving the modelling of a particular technology (reservoir hydropower), by adding constraints related to the physical nature of the phenomenon, never adopted in “pure” energy system models.

Lines 470-473: “Although these assumptions are very common in energy modelling practice [31], they still represent a limitation to the study, even though this limitation does not compromise the final aim of this work, which is adding constraints related to the physical nature of the reservoir hydropower to an energy system model.”

36. Line 416-418: How did you simulate hydropower plants outside ZRB? Are they simulated based on the very simplified approach that your paper criticizes? Please clarify and justify.

Yes this is correct, and we added clarification in the manuscript. The Zambezi River Basin was the test bench of this work, and hydropower falling outside of its boundaries were not taken into consideration in the new modelling approach.

Lines 482-484: “[…] those countries have no hydropower plants within the ZRB modelling scope, and have been hence modelled with Calliope_Base approach […]”

37. Figure 3: There are some strange patterns in this figure with regards to hydropower production. For example, there are rapid increases and drops in hydropower. Please explain why these happen. You need to provide the reader with details on reservoir operating rules to be able to interpret the results.

We assume the Reviewer means Figure 8 (now Figure 9). Thanks for highlighting this, a new paragraph has been inserted to explain such behaviours and give the readers all the necessary information.

Lines 550-556: “The reader can notice that results in Fig 9 presents some abrupt changes in hydropower production, specifically in subplots Mozambique Hydro 95-96 and Zimbabwe Hydro 95-96. Such abrupt changes are due to the nature of the energy model’s optimization nature, the objective function is the minimization of the Net Present Cost of the entire system, and in seeking this objective it makes use of the dams’ basins with this scope, putting economic benefits in front of dam’s operating rules. Indeed, better characterization of constraints of dam’s operating rules would help enhance further the effectiveness of this approach, that nonetheless already shows improvements with respect to the basic approach to the matter.”

38. Lines 491-492: This is simply incorrect. Many studies represented hydropower considering water availability constraints. The authors mention several examples of such studies. Please remove such erroneous claims.

Thanks for pointing out the mistake. We narrowed the claim to a more specific context, namely only power nodes energy system models widely in use.

Line 567-568: “Most widely adopted power nodes energy systems models supporting energy planning strategies represent hydropower reservoirs as dispatchable thermal plants”

39. Lines 500-505: Please mention that your model still shows a high bias compared to observed IEA data. This raises the question of how significant your claimed accuracy gains are compared to this high bias.

 We mentioned that in the manuscript. We would like to stress though that having a perfect representation of historical behaviour of the electricity dispatch strategy was not the aim of this work, which was instead, improving the way a power nodes energy model represents reservoir hydropower technology.

Lines 578: “even though still showing discrepancies from observed IEA data.”

40. Line 512-514: This should be mentioned earlier in the methodology. It is still unclear how dam operating rules are implemented in your model and how non-hydropower water users are simulated.

This was moved to methodology section.

Lines 324-333: “Spillage: due to the linear nature of the optimization framework of Calliope, it was not possible to accurately represent the activation of the dam spillways, i.e. releasing water out of the reservoir without whirling it in cases of excess of water to avoid dangerous overcharge of the dams. To better reproduce water spillages, it would be necessary to define a Boolean constraint dependent on the volume of water stored in the reservoir, which is activated when the storage exceeds a certain threshold. The proposed Calliope_Hydro instead models the spillage as a conversion technology connecting two cascade reservoirs with a conversion efficiency equal to zero. We made this choice to discourage the arbitrarily allocation of water from upstream to downstream reservoir according to an economical optimization. Setting the efficiency equal to zero, the spillage is modelled as a wasted water flow in order to be minimized by Calliope optimization.”

41. Lines 518-519: Again, this needs to be mentioned far earlier in the methodology. This is a major limitation. River systems are often not used for hydropower only.

This was moved to methodology section.

Lines 334-338: “Multi Stakeholder Management, as mentioned in the introduction of this work, is another critical aspect of water resource management, and consists in enlarging the scope of the analysis to include other water-related sectors beyond power production needs. This would require the formulation of other water demands alongside the electricity one, to account for competing uses of water in the river basin.” 

42. Generally, the article contains many language and grammar errors. I urge careful and perhaps professional proofreading.

Thanks for highlighting this. The manuscript has been carefully proofread before re-submitting.

REVIEWER #2 COMMENTS:

1. Line 16: Is the study now looking at hydropower generation in general or at “representation of storage hydropower” as mentioned in the highlights. Please clarify.

Thanks for highlighting the mismatch, we focus on reservoir hydropower. Sentence was corrected.

Line 17-18: “[…] advancing the representation of reservoir hydropower […]”

2. Line 23: The last sentence of the abstract is a bit misleading. Does this mean that the model is only applicable to Africa? Do only African reservoirs depend on water resources? In my opinion hydropower generates electricity but it cannot generate power.

The sentence was misleading, it was corrected to be valid in border terms.

Lines 25-27: “These improvements are useful to support hydropower management and planning capacity expansion in countries richly endowed with water resource or that are already strongly relying on hydropower for electricity production.”

3. Line 35: I would be careful with using the word “substantial” without any underlaying number.

The sentence was corrected.

Line 38: “Results suggest an improvement in the characterization of reservoir hydropower”

4. In my opinion the introduction to the water-nexus problematic false very short. Then the authors present a literature review (without specifying the criteria), but for me as reader it is unclear what of the information is relevant for the aim of the study, as this is only defined after the Literature review. But the literature review itself provides a very detailed overview. Thanks. But here one could argue that not all this information is needed to understand the aim of the study.

Thanks for highlighting that the entire section was unclear, we had reshaped the introduction, highlighting why we perform the literature review and what is the relevant information that we extract from that. Also, the water energy nexus introduction has been detailed more in depth, in order to let the reader identify the issue we aim to tackle with this work. 

5. Line 37: You are citing SDG 6 and SDG 7, but they have nothing per se to do with economics and poverty.

A more in-depth analysis of SDG6 and SDG7 relation with human and economic development has been performed.

Line 40-44: “Water [1] and Energy [2] are recognized by the United Nations as two of the 17 Sustainable Development Goals (SDGs) that humanity should pursue before 2030 for achieving sustainable development [3]. Water is a basic human right [4], no society can survive and prosper without it. On the other hand, energy is an instrumental human right, energy itself does not determine human dignity but with zero or poor access, fundamental rights may not be guaranteed [5].”

6. In my opinion Lines 37-40 are extremely short for a Water nexus introduction. I would suggest that the authors are a bit more specific which interconnection are relevant for this study.

Thanks for highlighting the shortcomings, this part has been expanded.

Lines 44-49: “Energy and water challenges are not independent and their interconnections, often entitled Water-Energy nexus [6,7], are increasingly recognized and studied [8–12]. “Water is needed for each stage of energy production, and energy is crucial for the provision and treatment of water” [7], and with the increase in needs for both energy and water worldwide, scientifically solid policies that regulate the energy sector and its water use and withdrawal without hindering energy security are needed to prevent future stress risk in particularly vulnerable areas.”

7. Line 40: If you are stating that the Water-Energy nexus is increasingly recognized and studied, it would be nice to provide a newer reference than 2015. I think a lot of research has been done since that.

Thanks for the suggestion, we agree and expanded the references accordingly.

Ref [8-12]

8. Line 42: Ref 5 Why are you referring to the Japanese translation?

We corrected it. Thanks for pointing out

9. Line 42: Of which nexus?

The sentence was expanded.

Line 51: “[…] key component of the water-energy nexus.”

10. Line 43: Largely? Wouldn’t that implement that it also works without water? I think its “inter alia” effected by climate variability and allocations to other uses, as for example also turbine efficiency and head can play a role for hydropower generation. Here again it would be nice to be more specific. What climate variability is relevant for hydropower and what are other users?

Increasingly more frequent and intense droughts and floods will impact hydropower production by inducing production curtailments to cope with water scarcity and head loss to buffer incoming floods. Most of hydropower dams are multipurpose and other uses include agriculture, industrial and domestic water supply, ecosystem preservation. We changed the sentence as follows: “climate variability (i.e. more frequent and intense droughts and floods) and allocations to other uses (e.g. irrigation, domestic and industrial water supply, ecosystem preservation)” Lines 52-54

11. Line 45: Please explain what you mean by “non-linear nature of hydropower”.

This sentence has been expanded to better explain what we mean by that in Lines 54-56:

“Despite the dynamic and non-linear nature of hydropower in phenomena such as evaporation losses and hydraulic head variation patters is well understood and considered in hydrological studies […]”

We would like to highlight as well that section 2.2 is completely dedicated to explaining this.

12. Line 59: Structure of the introduction: So far you have mentioned the research gap but not what the aim of the study is, but now I am presented with a chapter “Literature review”. Why has this been done, why is it important?

Thanks for the comment. The relevance of the literature review has been made explicit.

Lines 65-66: “An extensive review of literature concerning modelling energy and hydrological interdependencies related to hydropower is carried out in order to assess how past works delt with the issue.”

13. Line 71: You know mention “Reservoir hydropower”, before you talked about storage hydropower. Do these terms mean the same for you? I haven’t seen an explanation what you mean with storage hydropower yet.

We focus on Reservoir Hydropower, the nomenclature "storage hydropower" has been completely removed from the manuscript. Thanks for pointing out

14. Line 75. Here a reference for this statement should be added.

Thanks for the comment. We referenced the manual of the TIMES model adopted by the ETSAP programme of the IEA, as reference of an important and widely adopted energy model.

15. Line 88: Here you state that the energy model OsEMOSYS is enhanced. Why do you enhance this model?

This sentence was poorly written due to a typo, we do not deal with OSeMOSYS in this work, the sentence refers to previous work and has been corrected.

Line 99-101: “Another example is the “fix-and-relax” version of OSeMOSYS, mentioned in [38] and available at [39], in the mentioned work the energy model OSeMOSYS is enhanced under the hydropower representation point of view.”

16. Line 109: What do you mean by: “Literature is anyway rich of works”

The sentence was unclear and has reformulated:

Line 121-122: “We will now present works in which energy and hydrological models”

17. Line 169: Based on what criteria have you selected/ reviewed the literature? Is this overview meant to be comprehensive? More information is need.

 Thanks for highlighting this shortcoming, the required information have been added.

Lines 69-71: “The review of literature is conducted by selecting the most relevant works among the results of a query in Google Scholar under the keywords “Energy Modelling”, “Hydro Modelling”, “Reservoir Hydropower”, selecting works representing the six categories listed above.”

18. Line 174: Here you state that you are looking at “Calliope” but in line 88 you stated that you are enhancing the OsEMOSYS model. Please clarify.

The sentence highlighted by the Reviewer (line 88) was poorly written due to a typo and therefore was easily mis-understood, in this work we modify Calliope, not OSeMOSYS. The sentence has been corrected, thanks for pointing out. 

[See answer to Comment 15]

19. Line 186: What do you mean by energy carrier?

Citing from Wikipedia: An energy carrier is a substance (fuel) or sometimes a phenomenon (energy system) that contains energy that can be later converted to other forms such as mechanical work or heat or to operate chemical or physical processes. 

Examples: Electricity, Natural Gas, High Temperature Steam, etc.

https://en.wikipedia.org/wiki/Energy_carrier

https://www.sciencedirect.com/topics/engineering/energy-carrier

20. Line 193: Here you state that “40 GW of hydropower could be potentially deployed in this region”. But in the abstract, you write from a potential of 20,000 megawatts (MW). What is now correct?

In the abstract we refer to the Zambezi River Basin, in this sentence, to the entire SAPP. This is the reason behind this discrepancy.

21. Here the other show the used the Calliope model. However, in my opinion the authors fail to appropriately highlight what has been existing previously in the model and what their novel contribution is.

As I understand the other show in section 2.1 the Calliope_Base and then in Section 3.1 the Calliope_Hydro model. In the results you put the focus on the difference between Calliope_Base and by Calliope_Hydro. But the word Calliope_Hydro is for the first time mentioned in line 317. Maybe it would be better to combine he 2 chapters to highlight the differences?

Are run-off river power plants completely ignored by the model? Or are their now run-off river power plants in SAPP?

Section 2.1 defines the methodological strategy that transforms Calliope_Base into Calliope_Hydro, while section 3.1 describes the scope and assumptions of the Zambezi River Basin and SAPP case study implemented in the new Calliope_Hydro. 

Both sections were partially restructured and re-written to better highlight this.

Also Figure 2 reports the nomenclature Calliope_Hydro, as representation of the entire framework. 

For what concerns run-off river power plants: there are a few of them, but over all their installed capacity is negligible with respect to reservoir systems.

22. Line 204: I agree that factors are influencing the electricity production. But by how much do they influence the head? For me the head is the result of slope/high difference + water level in the reservoir. The factors you described are only looking at water level in the reservoir.

We meant that by changing the level the energy production is changed as it depends on the current hydraulic head (difference between the turbine axis elevation and the water level in the reservoir). We changed the sentence into “dynamic hydraulic head” to outline that we are referring to a time varying hydraulic head and not at the maximum hydraulic head which is a given feature of any dam.

Line 234.

23. Line 204: what do you mean by hydropower technologies? I think your factors are only relevant for reservoir hydropower but not for run-off-river hydropower.

The Reviewer is right, we had overlooked this mistake, the sentence has been corrected to the more specific term "Reservoir Hydropower". Line 235. 

24. Line 243: “that flows thought» wrong word?

It was a typo, thanks for pointing out. Now it is corrected to “through”. Line 277

25. In my opinion not only the amount of electricity potential, but also location and sizes of reservoirs should be presented. I am still lacking an explanation why the SAPP region has been selected. The authors then give a lot of explanation about the power pools. But wouldn’t be the type of hydropower and typical hydropower operations schemes, turbine/dam types more relevant for the reader?

Thanks for the comment, we have inserted Table 2 to cover this range of information.

26. Line 307: So there are no environmental regulations / minimum flow regulations that the hydropower operators have to balance as well?

There are minimum flow regulations only in ITT, but was not implemented as out of the scope of this work.

27. Line 307: By “adequate” you mean low water level?

We meant non-dangerous water levels in case of flood. The sentence has been expanded in order to be more clear.

Line 362-363: “ensuring adequate, or low enough, reservoir storage volume in order to avoid risks from incoming floods.”

28. Line 309: What is a large reservoir? Until now no information about reservoir size was provided. Does the “the mean annual river flow” refer to the river section in which the reservoir is located?

A large dam is a dam higher than 15 m as defined by the World Commission on Dams.

29. Line 311: Only downstream? What about the biodiversity in the lake?

Correct, we changed into “impact both in-reservoir and downstream ecosystem functions”.

Line 366 

30. Line 313: Please be consistent with the digits: 6.345 GW VS 4.91 GW

Thanks for pointing this out. It has been corrected.

31. Line 348: Reference to the map source is missing.

Thanks, the map has been substituted with an open-source one and has been referenced.

32. Line 375: From where do you receive your hydrological data?

Reporting from Line 365 (now line 425):

“Inflow data from 1986 to 2005 are extracted from the ADAPT project” [1] using the following gauging stations: Kafue Hook Bridge, Victoria Falls IN, Great East Road Bridge, and Mangochi. [1] J. Matos, A. Schleiss, J. Mertens and B. Wehrli, "Developing an open-source database for the Zambezi river basin," in Water Storage and Hydropower Development for Africa, Marrakesh (Morocco), 2015.

33. Line 383: Would be nice to get a chart of the river discharge?

We agree, Figure 6 has been added.

Figure 6 Cumulated historical inflow from 1986 to 2005 for Zambesi River Basin reservoirs

Line 446.

34. The results are presented in a nice and detailed way. However, I am the “discussion” part could be improved. For example: How would the results change if different time periods would have been chosen? What’s the difference in uncertanty between the Calliope Base and by Calliope_Hydro? In the abstract you highlighted that your support hydropower management and planning capacity in Africa. Please make the link to this statement. What is needed to apply you modell to other areas in Africa? Or globally?

Thanks for the comment. The Supplementary Information of the presented work contain all the outputs from the other scenarios considered, we feel like also adding such results in the main manuscript would make it heavier to read, being already very long. In addition to that and in accordance to the Reviewer’s comment we added Lines 561-564 to define what would be needed to replicate the study in other contexts: 

“In order to replicate the proposed approach to other large cascade reservoir basins, both in Africa or worldwide, it would only be necessary to know the geography of the basin, meaning the interconnections between the dams, the characteristics of the considered dams, as listed in Table 2 for this work, and the inflows expected for the modelling period.” 

35. You now showed in a case-study that its possible to move from Calliope_Base to Calliope_Hydro. But what’s the implication for the Energy Systems Modelling community?

 Many thanks for the very insightful comment, we gladly inserted lines 591-598 to give our perspective on the potential contributions of the proposed advancement to the energy system modelling community.

“The experiment presented in this work was carried out in the energy modelling framework Calliope, a power nodes model widely known and adopted in the open-energy-modelling community [31,32,56,57,59], but the issue of how to better represent reservoir hydropower is cross-cutting to the community and every linear optimization based power nodes model could benefit from this advancement implementing the same architecture in their framework. What is more, the proposed advancement, together with another previous work from the authors [65] - where heat pumps were modelled following a similar logic - suggests that the process of external iteration (Fig 3) could be a potential good practice to preserve linearity in energy system modelling.”

---

## [Decision Letter · Decision Letter 1]

8 Apr 2021

PONE-D-20-40043R1

Advancing the representation of Reservoir Hydropower in Energy Systems Modelling: the case of Zambesi River Basin

PLOS ONE

Dear Dr. Stevanato,

Thank you for submitting your manuscript to PLOS ONE. After careful consideration, we feel that it has merit but does not fully meet PLOS ONE’s publication criteria as it currently stands. Therefore, we invite you to submit a revised version of the manuscript that addresses the points raised during the review process.

Note that we still expect a MAJOR revision. Some of the previous comments remain major concerns. It is also important that comments are not only addressed in a response to the reviewers, but that the manuscript is revised accordingly.

We look forward to receiving your revised manuscript.

Kind regards,

Laura Scherer

Academic Editor

PLOS ONE

Journal Requirements:

Additional Editor Comments (if provided):

As both reviewers stress, the literature review is still not well written. It requires a major rewrite.

Likewise, the contribution of your study remains unclear. It even seems more unclear than before, as the scope of the study was clarified to be much more limited than it seemed to be.

Make assumptions and what is out of scope transparent and critically discuss the limitations of your study.

As pointed out earlier, let a native speaker carefully proofread the manuscript.

Reviewers' comments:

Reviewer's Responses to Questions

**Comments to the Author**

1. If the authors have adequately addressed your comments raised in a previous round of review and you feel that this manuscript is now acceptable for publication, you may indicate that here to bypass the “Comments to the Author” section, enter your conflict of interest statement in the “Confidential to Editor” section, and submit your "Accept" recommendation.

Reviewer #1: (No Response)

Reviewer #2: (No Response)

2. Is the manuscript technically sound, and do the data support the conclusions?

Reviewer #1: Partly

Reviewer #2: Partly

3. Has the statistical analysis been performed appropriately and rigorously? 

Reviewer #1: N/A

Reviewer #2: Yes

4. Have the authors made all data underlying the findings in their manuscript fully available?

Reviewer #1: Yes

Reviewer #2: Yes

5. Is the manuscript presented in an intelligible fashion and written in standard English?

Reviewer #1: No

Reviewer #2: Yes

6. Review Comments to the Author

Reviewer #1: Thank you to the authors for their effort to address my previous comments. Although the current version of the manuscript has improved over the previous version, it still needs to be improved further before publication. At this stage, I have two major concerns.

First, the literature review section is still not well written, and the methodological contribution of the paper does not fit well with how the literature review is narrated. The authors changed the main contribution of the paper to "improving hydropower representation in energy models." That is fine, but why is this at all important if other previous studies used integrated water-energy models that provide spatially and temporally explicit representation of hydropower and other water uses? I am missing this link. This might be because of how the literature review is structured.

My second major concern is the quality of language and grammar. The article is not well-written and contains numerous language and grammar errors. The authors should put more effort into proofreading the article or perhaps use a professional service provider.

Below are more comments on specific points:

1. Title: Please use small letters with "reservoir," "hydropower," "energy," "systems," and "modelling."

2. Line 21: since you modified the scope of the paper contribution, please add the word "energy" before "modelling framework."

3. Why is the literature review section number 1.1 and not 2, while the introduction is 1. This is not clear.

4. The literature review section is still very confusing. The authors attempted to address my previous comment by adding a paragraph to the beginning of the section. This is good, but not enough. The section needs a full rewrite. Breaking down the section into sub-sections might help remove some of the confusion.

5. Table 1: what is a "New model" integration?

6. Aim of the work: the contribution is still ambiguous. Why don't the authors explicitly state which open-source model they are trying to improve instead of saying, "improving the representation of reservoir hydropower within open-source energy systems modelling" and "improving the way reservoir hydropower is represented in existing energy system models." You need to be clear here that you are improving a certain open-source modelling tool.

7. Aim of the work: the scope of the work is to improve the representation of hydropower in an open-source energy modelling tool. That is fine, but how does this compare to integrated water-energy modeling approaches? Should the science and practice drop the idea of integrated water-energy modeling and try to improve hydropower representation in energy models? What I am missing here is a bridge between the literature review and the study contribution. It would be best to highlight how your work provides added value compared to previous approaches, including integrated water-energy modeling. This is a major issue that needs to be addressed. If the authors cannot find an added value of their proposed approach compared to comprehensive water-energy modeling, they should focus the paper around the case study rather than the modeling approach.

8. The issue of aggregated country-level energy demand nodes needs to be highlighted as a limitation, and its implications for the results need to be discussed.

9. Water use for purposes other than hydropower is not included in the case study analysis. This is a major issue. It would be helpful if the authors could attempt to add these other uses. Or at least discuss this limitation in more detail, including its implications for the presented results.

10. Previous comment number 34: the response does not make sense. The authors use historical data to validate the results of their approach. However, their SAPP model assumes a single market that did not exist in that historical period. I am failing to understand the logic here, but maybe I am missing something.

Reviewer #2: Dear authors,

Thanks for improving the manuscript. Despite your big effort in the revision I still have some comments:

Generall comment:

The authors limited the scope of the study dramatically in the new version: "The contribution we propose is the enhancement of an energy system model alone: no integration between energy and hydro models is performed, thus framing our contribution in a different context." This makes it even more imporant to hilglight the novelty of the study, since its only apllied to one basin.

Further the other state 586-588: "In order to replicate the proposed approach to other large cascade reservoir basins, both in Africa or worldwide, it would only be necessary to know the geography of the basin, meaning the interconnections between the dams, the characteristics of the considered dams, as listed in Table 2 for this work, and the inflows expected for the modelling period."

 As it seems that data is availabel for Africa, I wounder why the manuscript is limited to such a small case-study.

REVIEWER#2 COMMENTS:

10: Author comment :We changed the sentence as follows: “climate variability (i.e. more frequent and intense droughts and floods) and allocations to other uses (e.g. irrigation, domestic and industrial water supply, ecosystem preservation)” Lines 52-54

Reviewer reply (R) : so in the end climate variability means precipitation? Does an increase in precipitation always directly lead to more floods?

12: Author comment : Line 59: Structure of the introduction: So far you have mentioned the research gap but not what the aim of the study is, but now I am presented with a chapter “Literature review”. Why has this been done, why is it important? Thanks for the comment. The relevance of the literature review has been made explicit. Lines 65-66: “An extensive review of literature concerning modelling energy and hydrological interdependencies related to hydropower is carried out in order to assess how past works delt with the issue.”

R: I feel sorry to say that, but for me personally the introduction section structure could still be improved further.

In section 1 you end with what current models do not account for.

and in section 1.1 you start with “delt with the issue”

But what I think is mentioned is: what is the consequence or issue of the research gap presented in section 1?

17: Author comment : Line 169: Based on what criteria have you selected/ reviewed the literature? Is this overview meant to be comprehensive? More information is need Thanks for highlighting this shortcoming, the required information have been added. Lines 69-71: “The review of literature is conducted by selecting the most relevant works among the results of a query in Google Scholar under the keywords “Energy Modelling”, “Hydro Modelling”, “Reservoir Hydropower”, selecting works representing the six categories listed above

R: So you used “” which, retrieves an exact match of phrase, without any wildcard. So what about : Modelling (UK spelling) vs modeling (Us spelling). So you only searched for for papers with UK spelling? With “reservoir hydropower” you will not find “model for hydropower reservoirs”

How many papers have you assessed to get Table 1?

You now just simply added the 2 publications mentioned by reviewer 1 in a new category. That contradicts your methods of the literature review. Would you find them with the used key-words? Or would that require a change of key-words?

I would like to refer to Reviewer 1, Comment 1, and also ask Reviewer 1 once more to check for comprehensives of the Literature review, considering the search key-words.

In lines 65-99 you introduce 6 model categories. Wouldn’t it be good to refer to the same categories in Table 1?

21: Author comment: Section 2.1 defines the methodological strategy that transforms Calliope_Base intoCalliope_Hydro, while section 3.1 describes the scope and assumptions of theZambezi River Basin and SAPP case study implemented in the new Calliope_Hydro.Both sections were partially restructured and re-written to better highlight this.

R: According to the track changes you added 1 sentence in section 2.1 and 2 in section 3.1. Can we consider this as restructured and re-written?

21. Author comment :A: For what concerns run-off river power plants: there are a few of them, but over all their installed capacity is negligible with respect to reservoir system

R:Maybe it would be good to state that in the manuscript and not inly in the reviewer reply?

25: Author comment : In my opinion not only the amount of electricity potential, but also location and sizes of reservoirs should be presented. I am still lacking an explanation why the SAPP region has been selected. The authors then give a lot of explanation about the power pools. But wouldn’t be the type of hydropower and typical hydropower operation schemes, turbine/dam types more relevant for the reader

R: Thanks for adding the table, that help a lot. But I am still missing an explanation why the SAPP region has been selected or why the other thing that it is a good area for a case study? Or was it simply chosen due to data availability? That’s also fine, but should be mentioned.

26: Author comment: Line 307: So there are no environmental regulations / minimum flow regulations that the hydropower operators have to balance as well? There are minimum flow regulations only in ITT, but was not implemented as out of the scope of this work

R: Then I think it should be mentioned that they are out of scope.

28 Author comment: Line 309: What is a large reservoir? Until now no information about reservoir size was provided. Does the “the mean annual river flow” refer to the river section in which the reservoir is located? A large dam is a dam higher than 15 m as defined by the World Commission on Dam

R: How should the reader now that you refer to the definition by the “World Commission on Dam”

32: Author comment: Thanks for point out that you provided the hydrological data. But maybe then state it in the manuscript as you did in the reviewer reply. “ “Inflow data from 1986 to 2005 are extracted from the ADAPT project”[1] using the following gauging stations: Kafue Hook Bridge, Victoria Falls IN, Great East Road Bridge, and Mangochi. [1]» As for now in Line 425 it only says: Inflow data(the SW block in Figure 5) are extracted from the ADAPT project [87].

7. PLOS authors have the option to publish the peer review history of their article (what does this mean?). If published, this will include your full peer review and any attached files.

Reviewer #1: No

Reviewer #2: No

---

## [Author Response · Author response to Decision Letter 1]

16 Jun 2021

EDITOR COMMENTS:

1. As both reviewers stress, the literature review is still not well written. It requires a major rewrite.

The literature review has been substantially restructured and re-written. We hope it now meets the quality standards required.

2. Likewise, the contribution of your study remains unclear. It even seems more unclear than before, as the scope of the study was clarified to be much more limited than it seemed to be.

We hope that by improving the literature review and changing the wording of the “aim of the work” section, this is now clearer.

3. Make assumptions and what is out of scope transparent and critically discuss the limitations of your study.

Following Reviewers’ comments we stated every assumption and limitation of the work.

4. As pointed out earlier, let a native speaker carefully proofread the manuscript.

We have increased our efforts to improve the English throughout the manuscript. But the Editor still feels that we should pursue professional proofreading we can do so.

REVIEWER #1 COMMENTS:

1. Title: Please use small letters with "reservoir," "hydropower," "energy," "systems," and "modelling."

It has been corrected. 

2. Line 21: since you modified the scope of the paper contribution, please add the word "energy" before "modelling framework".

Thanks for pointing out. It has been inserted.

3. Why is the literature review section number 1.1 and not 2, while the introduction is 1. This is not clear.

Please consider that the aim of the work is not the literature review, for this reason the review of previous works on the matter was considered to be part of the introduction, as a way to frame the intervention in the broader picture. For this reason, the literature review was not dedicated its own chapter. For the same reason, and according to the following comments and comments form Reviewer #2, we have changed name and scope of the subchapters, without conducting a full-fledged literature review. We restructured chapter 1 as follows:

1.1 Energy and Water

1.2 Modelling hydrological and energy systems

1.3 Representation of hydropower in energy modelling

1.4 Aim of the work

We believe this way the scope of the work is better framed and justified.

4. The literature review section is still very confusing. The authors attempted to address my previous comment by adding a paragraph to the beginning of the section. This is good, but not enough. The section needs a full rewrite. Breaking down the section into sub-sections might help remove some of the confusion.

This comment helped in reshaping Chapter 1, see answer to previous comment for details.

5. Table 1: what is a "New model" integration?

The term “New Model” was meant to identify an alternative to the integration of two models, being in fact the creation of a new model from scratch. We decided to remove Table 1 from the manuscript as it was drawing too much attention on the previous modelling efforts, while the aim of the work is not to conduct a literature review. The identification of previous efforts is only instrumental to provide the background knowledge to the reader to fully capture the value of our proposed approach. 

6. Aim of the work: the contribution is still ambiguous. Why don't the authors explicitly state which open-source model they are trying to improve instead of saying, "improving the representation of reservoir hydropower within open-source energy systems modelling" and "improving the way reservoir hydropower is represented in existing energy system models." You need to be clear here that you are improving a certain open-source modelling tool.

In line 217 we state that the model we are using as proof of concept is Calliope, but the validity of the approach is wider that the single case of Calliope. Indeed, every energy modelling framework defined based on power nodes (1) could potentially benefit from the proposed approach.

(1) (Heussen, K. et al., 2010. Energy storage in power system operation: The power nodes modelling framework. In Innovative Smart Grid Technologies Conference Europe (ISGT Europe), 2010 IEEE PES. pp. 1–8. DOI: 10.1109/ISGTEUROPE.2010.5638865).

7. Aim of the work: the scope of the work is to improve the representation of hydropower in an open-source energy modelling tool. That is fine, but how does this compare to integrated water-energy modeling approaches? Should the science and practice drop the idea of integrated water-energy modeling and try to improve hydropower representation in energy models? What I am missing here is a bridge between the literature review and the study contribution. It would be best to highlight how your work provides added value compared to previous approaches, including integrated water-energy modeling. This is a major issue that needs to be addressed. If the authors cannot find an added value of their proposed approach compared to comprehensive water-energy modeling, they should focus the paper around the case study rather than the modeling approach.

Thanks for the comment, it allowed us to better frame our contribution: 

Lines 138-150: “The issue with the presented cases is the lack of replicability of those integrated energy-water frameworks. The mentioned approaches are indeed rarely replicated to other geographical areas, given the complex structure of the integrated models, which are sometimes ad-hoc developed around a specific case study. The scientific value of such integrated models is indeed relevant and not questioned in this work, which is instead focused on model replicability in broader engineering applications. 

On the other hand, most energy system models (both proprietary and open-source) are adopted as a standard by the international community [35] and used in a wide range of applications, as discussed later in section ‘Aim of the work’. However, their representation of a variety of engineering phenomena, including hydropower, leave space for improvements. Therefore, improving an already existing energy modelling framework may have a broader impact on the international community in terms of model accessibility by users and replicability of the analysis in other contexts.

Given these considerations, this work proposes a methodology for improving the reservoir hydropower representation in open-source energy system models.”

8. The issue of aggregated country-level energy demand nodes needs to be highlighted as a limitation, and its implications for the results need to be discussed.

In modelling energy systems, the level of aggregation of country energy demand depends by the aim and scope of the analysis and, secondly, the availability of detailed energy demand data. 

For example, if the analysis aims to investigate the reliability of transmission and distribution network within a country, then a single-node model would be not appropriate. The focus of the proposed application is instead in quantifying the effects of a better characterization of the hydropower system in a multi-regional context, assessing the changes in national supply technology mixes. 

Given the complexity of national energy systems and the non-homogenous distribution of natural resources and final energy demand, relying on multi-nodal representation of each country may provide more detailed and accurate results. However, the paucity and high uncertainty of raw energy data in the sub-Saharan region must be properly considered in order to avoid that on overly detailed model relying low quality data results in an uncontrolled error propagation.

Finally, please consider the following relevant and recent literature references, adopting single-region energy models applied in the African context.

Pavičević, De Felice, Busch, Hidalgo González, Quoilin, Water-energy nexus in African power pools - The Dispa-SET Africa model, Energy, Volume 228, 2021, 120623, https://doi.org/10.1016/j.energy.2021.120623

Taliotis, Shivakumar, Ramos, Howells, Mentis, Sridharan, Broad, Mofor, An indicative analysis of investment opportunities in the African electricity supply sector — Using TEMBA (The Electricity Model Base for Africa), Energy for Sustainable Development, Volume 31, 2016, Pages 50-66, https://doi.org/10.1016/j.esd.2015.12.001

In light of this, Lines 465-472 were added: “A further limitation of the modelling approach is the single node representation of the energy demand of each country, exception made for Mozambique, characterized by two demand nodes. In energy system modelling, the level of aggregation of country energy demand depends by the aim and scope of the analysis and, secondly, the availability of detailed energy demand data. Increasing the number of demand nodes characterization may have resulted in more detailed and accurate results. For the context of analysis, one demand node per country was considered a fair trade-off between the data paucity that characterizes the Sub-Saharan context and results accuracy. This choice is anyway in line with the most recent literature on the subject [72,73].”

9. Water use for purposes other than hydropower is not included in the case study analysis. This is a major issue. It would be helpful if the authors could attempt to add these other uses. Or at least discuss this limitation in more detail, including its implications for the presented results.

We agree that the use of water for other purposes is an important aspect, and it was in fact included in the “Current Limitations” section. We discussed the implications of this limitation further in the manuscript.

Line 369-375: “Not considering the non-hydropower uses of water implies that the amount of water that appears to be available for the turbines according to the model is actually more than the amount available in real life. The main other use of water in the area is water for agriculture, that is extracted directly from the river basin and from the dams, the water used for agriculture should hence be subtracted from the amount of water flowing in the system. This could be modelled in the framework by inserting, as stated above, a demand for non-hydropower water, that has to be satisfied by the system; that water would then be removed from the system.”

Line 614-616: “It is worth noting that including in the analysis also non-hydropower uses of water, would shift even more in this results towards a lower availability of hydropower in the system, given the fact that considering it would decrease the amount of water available in the system for power production purposes.”

10. Previous comment number 34: the response does not make sense. The authors use historical data to validate the results of their approach. However, their SAPP model assumes a single market that did not exist in that historical period. I am failing to understand the logic here, but maybe I am missing something.

To test the effects that different water inflow scenarios have on energy supply dispatch, we selected real water inflow scenarios, ranging from wetter to dryer, based on empirically observed data in different past years (some of which are 20 years old). We selected those data with the sole purpose of relying on realistic inflow patterns.

As specified in the reply to your past comment #34, the aim of this case study is to test the change in behaviour of the model according to an improved modelling approach for the hydropower system, relying on different hydrological inflow profiles. The nature of the power system, either single or open energy market is not relevant for the current application, nor hindering the validity of the obtained results. This is in accordance with the common practice in energy system modelling for multi-regional cases, where the model works based on a single, perfectly informed energy regulatory authority, which minimizes the overall operational costs of energy generation considering the entire area as a single market.

REVIEWER #2 COMMENTS:

General: As it seems that data is available for Africa, I wounder why the manuscript is limited to such a small case-study.

Because it was interesting to focus on the inter-national dynamics of electricity exchange in a basin of sub-continental relevance, enlarging the analysis to other basins would not add anything to the methodology, the main contribution of the work; the case study is only meant to show an application. In addition, Africa is not all interconnected, the analysis of the entire continent would just result in a series of smaller isolated sub-continental power pools, like the SAPP analysed in this case.

10. Author comment :We changed the sentence as follows: “climate variability (i.e. more frequent and intense droughts and floods) and allocations to other uses (e.g. irrigation, domestic and industrial water supply, ecosystem preservation)” Lines 52-54

Reviewer reply (R) : so in the end climate variability means precipitation? Does an increase in precipitation always directly lead to more floods?

The sentence has been expanded into: 

“Hydropower generation is largely dependent upon climate variability, which may either curtail production during intense drought events that reduce water availability [14] or induce large water losses as a consequence of spilling during flood events [15]. In addition, hydropower production is also influenced by other competing water users such as irrigation, domestic and industrial water supply, and ecosystem preservation (e.g., 40% of existing hydropower dams serve multiple demands [16])” Lines: 53-57

12. Author comment : Line 59: Structure of the introduction: So far you have mentioned the research gap but not what the aim of the study is, but now I am presented with a chapter “Literature review”. Why has this been done, why is it important? Thanks for the comment. The relevance of the literature review has been made explicit. Lines 65-66: “An extensive review of literature concerning modelling energy and hydrological interdependencies related to hydropower is carried out in order to assess how past works delt with the issue.”

R: I feel sorry to say that, but for me personally the introduction section structure could still be improved further.

In section 1 you end with what current models do not account for.

and in section 1.1 you start with “delt with the issue”

But what I think is mentioned is: what is the consequence or issue of the research gap presented in section 1?

The entire Chapter 1 has been restructured, expanded and overall improved to better frame our intervention in the background identified, according to this comment, following comments on literature review and Reviewer #1 comments on the subject.

17. Author comment : Line 169: Based on what criteria have you selected/ reviewed the literature? Is this overview meant to be comprehensive? More information is need Thanks for highlighting this shortcoming, the required information have been added. Lines 69-71: “The review of literature is conducted by selecting the most relevant works among the results of a query in Google Scholar under the keywords “Energy Modelling”, “Hydro Modelling”, “Reservoir Hydropower”, selecting works representing the six categories listed above

R: So you used “” which, retrieves an exact match of phrase, without any wildcard. So what about : Modelling (UK spelling) vs modeling (Us spelling). So you only searched for for papers with UK spelling? With “reservoir hydropower” you will not find “model for hydropower reservoirs”

How many papers have you assessed to get Table 1?

You now just simply added the 2 publications mentioned by reviewer 1 in a new category. That contradicts your methods of the literature review. Would you find them with the used key-words? Or would that require a change of key-words?

I would like to refer to Reviewer 1, Comment 1, and also ask Reviewer 1 once more to check for comprehensives of the Literature review, considering the search key-words.

In lines 65-99 you introduce 6 model categories. Wouldn’t it be good to refer to the same categories in Table 1?

The entire structure of chapter 1 has been changed, and the reviewed publications do not constitute a “Literature Review” anymore, but are just used to frame our intervention in a background that justifies the relevance of the approach. Please consider that a literature review is not the goal of the paper, and the attention of the reader should be focused on the methodology. 

Table 1 does has been removed as it was not relevant to introduce the aim of the work. 

21a. Author comment: Section 2.1 defines the methodological strategy that transforms Calliope_Base into Calliope_Hydro, while section 3.1 describes the scope and assumptions of the Zambezi River Basin and SAPP case study implemented in the new Calliope_Hydro. Both sections were partially restructured and re-written to better highlight this.

R: According to the track changes you added 1 sentence in section 2.1 and 2 in section 3.1. Can we consider this as restructured and re-written?

21b. Author comment :A: For what concerns run-off river power plants: there are a few of them, but over all their installed capacity is negligible with respect to reservoir system

R:Maybe it would be good to state that in the manuscript and not inly in the reviewer reply?

21a. It seems from this comment that is not clear from the text what was present in Calliope_Base and what has been enhanced in Calliope_Hydro, and in which sections of the manuscript this has been explained.

We added the following lines to make this more clear:

Line 274-277: “Section 2.1 and 2.2 describe the modelling methodology adopted for enhancing Calliope_Base into Calliope_Hydro, taking into consideration the four points highlighted at the beginning of this chapter. Section 3 presents the case study and how the new methodology is applied to it, while section 4 presents and comments the results.”

Line 279-284: “In the current version of Calliope, here referred to as Calliope_Base, the modeler can make use of a series of predefined supply and conversion technologies archetypes (i.e. Supply, Supply +, Conversion, Conversion +, Storage) and the standard way of modelling reservoir hydropower is by means of a Supply + technology, a technology that allows a flow of energy to enter the system based on a determined resource availability, and that can have a storage integrated, (like for example a PV power plant with a Battery Energy Storage System). Such existing technology archetypes are used here to better model reservoir hydropower.”

Line 285-286: “In this subsection we outline how we delt with points (I) time-dependent water inflow patterns in the basin due to hydrological processes and (II) water inflow supplied by upstream reservoirs.”

Line 317-318: “In this subsection we outline how we delt with points (III) dam maximum and minimum operational level and related water release constraints and (IV) evaporation losses dependent upon the reservoir surface.”

21b. We agree, it has been inserted. Lines 406-408: “Run-off river plants exist in the considered study area, but their installed capacity is negligible with respect to the considered reservoir system, and for this reason left out of the scope of the modelling effort carried out in this work.”

25. Author comment : In my opinion not only the amount of electricity potential, but also location and sizes of reservoirs should be presented. I am still lacking an explanation why the SAPP region has been selected. The authors then give a lot of explanation about the power pools. But wouldn’t be the type of hydropower and typical hydropower operation schemes, turbine/dam types more relevant for the reader

R: Thanks for adding the table, that help a lot. But I am still missing an explanation why the SAPP region has been selected or why the other thing that it is a good area for a case study? Or was it simply chosen due to data availability? That’s also fine, but should be mentioned.

Line 408-414: “The ZRB and the SAPP were selected as case study because they represent an extremely suitable ground for testing our approach, different countries with different electricity generation mix, interconnected with each other and with a relevant river basin flowing through many of the involved countries, with dams distributed among them. It is an area widely studied by the scientific community in terms of Water-Energy Nexus [78] and in addition to that, the participation of the authors in the DAFNE project (https://dafne.ethz.ch/) made data easily accessible making in it the final decision for the case study.”

26. Author comment: Line 307: So there are no environmental regulations / minimum flow regulations that the hydropower operators have to balance as well? There are minimum flow regulations only in ITT, but was not implemented as out of the scope of this work

R: Then I think it should be mentioned that they are out of scope.

Correct, we inserted it. Line 486-487: “For what concerns environmental and minimum flow regulations, only ITT has minimum flow regulations, but they were not implemented as out of the scope of this work.”

28. Author comment: Line 309: What is a large reservoir? Until now no information about reservoir size was provided. Does the “the mean annual river flow” refer to the river section in which the reservoir is located? A large dam is a dam higher than 15 m as defined by the World Commission on Dam

R: How should the reader now that you refer to the definition by the “World Commission on Dam”

Line 477: “The four dams are considered “large dams” according to the World Commission of Dams [96], being higher than 15 m.”

32. Author comment: Thanks for point out that you provided the hydrological data. But maybe then state it in the manuscript as you did in the reviewer reply. “ “Inflow data from 1986 to 2005 are extracted from the ADAPT project”[1] using the following gauging stations: Kafue Hook Bridge, Victoria Falls IN, Great East Road Bridge, and Mangochi. [1]» As for now in Line 425 it only says: Inflow data(the SW block in Figure 5) are extracted from the ADAPT project [87].

Lines 477-481: “Table 2 reports details on the four dams. Notably, the dams are located in different countries and may be linked with each other. Inflow data (the SW block in Figure 5) from 1986 to 2005 are extracted from the ADAPT project [97] using the following gauging stations: Kafue Hook Bridge, Victoria Falls IN, Great East Road Bridge, and Mangochi.”

---

## [Decision Letter · Decision Letter 2]

3 Aug 2021

PONE-D-20-40043R2

Advancing the representation of Reservoir Hydropower in Energy Systems Modelling: the case of Zambesi River Basin

PLOS ONE

Dear Dr. Stevanato,

Thank you for submitting your manuscript to PLOS ONE. After careful consideration, we feel that it has merit but does not fully meet PLOS ONE’s publication criteria as it currently stands. Therefore, we invite you to submit a revised version of the manuscript that addresses the points raised during the review process.

I agree with the two reviewers that the manuscript greatly improved. Thank you for the revision. Some areas of improvement still remain, as you will see based on the comments below. Most importantly, please address the limitation of assuming a single market in your discussion.

Reviewer 2 suggests to shorten the introduction, as it is longer than the methods described in section 2. Although I agree that the introduction is quite long and could be shortened, I recognize that section 3 about the case study also describes methods and the methods description is, thus, longer than perceived by that reviewer. I suggest to only shorten the introduction to some extent if it doesn’t change the content anymore, as the introduction has already undergone a major revision in previous rounds.

We look forward to receiving your revised manuscript.

Kind regards,

Laura Scherer

Academic Editor

PLOS ONE

Journal Requirements:

Additional Editor Comments:

In Table 1, please indicate the order of the latitude and longitude for the coordinates column.

In Figures 6 to 9, please remove the figure titles. They are redundant, given the figure captions.

Please deposit your data and code in a static repository with a doi. GitHub is not a proper repository for such a purpose. You can, for example, use Zenodo to archive a GitHub repository, make it static, and assign a doi. You can find further recommended repositories here: https://journals.plos.org/plosone/s/recommended-repositories

Reviewers' comments:

Reviewer's Responses to Questions

**Comments to the Author**

1. If the authors have adequately addressed your comments raised in a previous round of review and you feel that this manuscript is now acceptable for publication, you may indicate that here to bypass the “Comments to the Author” section, enter your conflict of interest statement in the “Confidential to Editor” section, and submit your "Accept" recommendation.

Reviewer #1: (No Response)

Reviewer #2: All comments have been addressed

2. Is the manuscript technically sound, and do the data support the conclusions?

Reviewer #1: Partly

Reviewer #2: Yes

3. Has the statistical analysis been performed appropriately and rigorously? 

Reviewer #1: Yes

Reviewer #2: Yes

4. Have the authors made all data underlying the findings in their manuscript fully available?

Reviewer #1: Yes

Reviewer #2: Yes

5. Is the manuscript presented in an intelligible fashion and written in standard English?

Reviewer #1: No

Reviewer #2: Yes

6. Review Comments to the Author

Reviewer #1: I thank the authors for their effort to address my previous comments. The current version of the manuscript improved markedly and can be accepted for publication after addressing the following two issues:

*As I stated in my previous reports, the article requires professional proofreading before it can be published.

*Your response number 10: the response still does not makes sense to me. The authors state, “The nature of the power system, either single or open energy market is not relevant for the current application, nor hindering the validity of the obtained results.” Most African countries are not interconnected as a single market, and simulating them as a single market where electricity generation and use is driven by costs and prices is unrealistic and will undoubtedly have implications for the results. The authors mention in line 525 that the simulation error is because of this market assumption; so, this assumption has implications for the results. Models should represent reality to an acceptable level, and getting the type of market right is necessary for the present context—this major limitation of the case study application needs to be highlighted to the reader more clearly. You should add new text to discuss the implications of this limitation.

Reviewer #2: Dear authors,

Thank you very much for submitting this very much improved version of the manuscript and considering most of my comments.

However, I still have some comments regarding the Introduction.

1:

R2: Thanks a lot for removing the Literature review section and adding the section “Representation of hydropower in energy modelling”. This section is much better. Thanks.

However, when it comes to the introduction , section “1.4. Aim of the work” still needs to be improved. Looking at the track changes I see only 2 small changes despite the comments for the editor:

“2. Likewise, the contribution of your study remains unclear. It even seems more unclear than before, as the scope of the study was clarified to be much more limited than it seemed to be”

and reviewer 1:

“7. Aim of the work: the scope of the work is to improve the representation of hydropower in an open-source energy modelling tool. That is fine, but how does this compare to integrated water-energy modeling approaches? Should the science and practice drop the idea of integrated water-energy modeling and try to improve hydropower representation in energy models? What I am missing here is a bridge between the literature review and the study contribution. It would be best to highlight how your work provides added value compared to previous approaches, including integrated water-energy modeling. This is a major issue that needs to be addressed. If the authors cannot find an added value of their proposed approach compared to comprehensive water-energy modeling, they should focus the paper around the case study rather than the modeling approach”

You mentioned that you changed something in lines 138-150. This is however not in the “Aim of the work chapter”. In other words, the link between 1.3 and 1.4 should still be improved.

On another note, the authors sate: “Please consider that a literature review is not the goal of the paper, and the attention of the reader should be focused on the methodology.”

However, Lines 39-244 are dedicated for the introduction, while lines 246 – 376 are for the methods.

In other words, the introduction is longer than the methods. If the focus of the reader should be the methodology, I would argue that the new intro is now unfortunately to long and could be shortened.

7. PLOS authors have the option to publish the peer review history of their article (what does this mean?). If published, this will include your full peer review and any attached files.

Reviewer #1: No

Reviewer #2: No

---

## [Author Response · Author response to Decision Letter 2]

20 Sep 2021

Ref: PONE-D-20-40043

Title: Advancing the representation of reservoir hydropower in energy systems modelling: the case of Zambesi River Basin

Journal: PLoS One 

Response to Reviewers:

EDITOR COMMENTS:

1. Reviewer 2 suggests to shorten the introduction, as it is longer than the methods described in section 2. Although I agree that the introduction is quite long and could be shortened, I recognize that section 3 about the case study also describes methods and the methods description is, thus, longer than perceived by that reviewer. I suggest to only shorten the introduction to some extent if it doesn’t change the content anymore, as the introduction has already undergone a major revision in previous rounds.

We agree that the introduction is quite long, we reduced the detail of the description of the literature reviewed in chapter 1.2, without hindering the sense or changing the content. Since, as agreed, the scope of the work is not a literature review, we believe this modification does not subtract to the overall quality and does not reduce the scope of the work. For what concerns chapter 1.3, we did not reduce the detail of description because we believe that it is instrumental to the introduction of the scope of the work to fully describe representation of hydropower in existing energy models.

2. In Table 1, please indicate the order of the latitude and longitude for the coordinates column.

This has been fixed.

3. In Figures 6 to 9, please remove the figure titles. They are redundant, given the figure captions.

This has been taken care of.

4. Please deposit your data and code in a static repository with a doi. GitHub is not a proper repository for such a purpose. You can, for example, use Zenodo to archive a GitHub repository, make it static, and assign a doi. 

Thanks for suggesting this. The github repository has been deposited in Zenodo and the relative doi is: https://doi.org/10.5281/zenodo.5242926

REVIEWER #1 COMMENTS:

1. As I stated in my previous reports, the article requires professional proofreading before it can be published

We did our best to further improve the language.

2. Your response number 10: the response still does not makes sense to me. The authors state, “The nature of the power system, either single or open energy market is not relevant for the current application, nor hindering the validity of the obtained results.” Most African countries are not interconnected as a single market, and simulating them as a single market where electricity generation and use is driven by costs and prices is unrealistic and will undoubtedly have implications for the results. The authors mention in line 525 that the simulation error is because of this market assumption; so, this assumption has implications for the results. Models should represent reality to an acceptable level, and getting the type of market right is necessary for the present context—this major limitation of the case study application needs to be highlighted to the reader more clearly. You should add new text to discuss the implications of this limitation.

This limitation has been highlighted and discussed. Lines 630-639.

“Another limitation of the proposed case study, even though common in similar studies [94,95], is the assumption of a single market for the entire power pool. Each of the modelled countries, even though exchanging with the neighboring ones, is not yet part of a single SAPP energy market. This assumption partially influences the results in terms of exchanged power, and in turn, of produced power. The model optimizes on the minimization of the overall cost of operating the entire system, in this way the energy mix of a country is influenced by its possibility to fulfill the energy needs of another country. This results in a higher use of technologies with lower operation costs, that are employed to cover the demand of the countries that have more costly technologies in their production park, inside the limits of transmission capacity. Nonetheless, this does not hinder the value of the proposed methodology, that is still demonstrated to be valid by the case study.”

REVIEWER #2 COMMENTS:

1: Thanks a lot for removing the Literature review section and adding the section “Representation of hydropower in energy modelling”. This section is much better. Thanks.

However, when it comes to the introduction, section “1.4. Aim of the work” still needs to be improved. Looking at the track changes I see only 2 small changes despite the comments for the editor: “2. Likewise, the contribution of your study remains unclear. It even seems more unclear than before, as the scope of the study was clarified to be much more limited than it seemed to be”, and reviewer 1: “7. Aim of the work: the scope of the work is to improve the representation of hydropower in an open-source energy modelling tool. That is fine, but how does this compare to integrated water-energy modeling approaches? Should the science and practice drop the idea of integrated water-energy modeling and try to improve hydropower representation in energy models? What I am missing here is a bridge between the literature review and the study contribution. It would be best to highlight how your work provides added value compared to previous approaches, including integrated water-energy modeling. This is a major issue that needs to be addressed. If the authors cannot find an added value of their proposed approach compared to comprehensive water-energy modeling, they should focus the paper around the case study rather than the modeling approach”

You mentioned that you changed something in lines 138-150. This is however not in the “Aim of the work chapter”. In other words, the link between 1.3 and 1.4 should still be improved.

We agree that was missing a connection between chapter 1.3 and 1.4, and chapter 1.4 was starting abrupted. We have inserted now lines 191-198:

“As observed in chapter 1.2 and 1.3, model integration between energy and hydro models results in drawbacks in terms of computational time and non-replicability, among others. On the other hand, while only-energy models, and in particular open-source ones, are meant to build in the direction of solving such issues, they lack in proper characterization of hydrological phenomena and hydropower production. For these reasons, the aim of this work is to provide a methodology for improving the characterization of reservoir hydropower in open-energy-modelling. Through this methodology we aim at reaching an advance in the energy modelling science, by learning from the integrated modelling experience analyzed in chapter 1.2, and overcoming the highlighted limits thanks to the advantages brough about by the open-modelling framework.”

And we believe this lines give now the proper structure to the chapter.

2. On another note, the authors sate: “Please consider that a literature review is not the goal of the paper, and the attention of the reader should be focused on the methodology.”

However, Lines 39-244 are dedicated for the introduction, while lines 246 – 376 are for the methods.

Thanks for highlighting this, measures for shortening the introduction have been taken. Please see answer to Editor’s comment #1 for a more detailed response.

---

## [Editor Report · Decision Letter 3]

1 Oct 2021

PONE-D-20-40043R3Advancing the representation of Reservoir Hydropower in Energy Systems Modelling: the case of Zambesi River BasinPLOS ONE

Dear Dr. Stevanato,

Thank you for submitting your manuscript to PLOS ONE. After careful consideration, we feel that it has merit but does not fully meet PLOS ONE’s publication criteria as it currently stands. Therefore, we invite you to submit a revised version of the manuscript that addresses the points raised during the review process. The manuscript is already at a stage where I do not need to send it back to reviewers. However, I would like to see some final editorial changes before the manuscript can be accepted for publication. First, all figures should be well readable; however, some font sizes need to be increased to achieve that. This applies especially to Figures 6 and 9 and to a lesser extent to Figures 7 and 8. Second, the manuscript requires some copyediting, at least with a tool like Grammarly (it has a free version). For example, I think whenever you use "whom", it should be "who".

We look forward to receiving your revised manuscript.

Kind regards,

Laura Scherer

Academic Editor

PLOS ONE
---

## [Author Response · Author response to Decision Letter 3]

14 Oct 2021

Dear Editor, 

Thanks for the positive comments, we are glad the manuscript now meets the scientific standards of your journal. We have increased the font of Figures 6 to 9, and a complete review of the manuscript with Grammarly has been performed.

We hope that this meets your demands.

Sincerely,

Nicolò Stevanato

---

## [Editor Report · Decision Letter 4]

29 Oct 2021

Advancing the representation of Reservoir Hydropower in Energy Systems Modelling: the case of Zambesi River Basin

PONE-D-20-40043R4

Dear Dr. Stevanato,

We’re pleased to inform you that your manuscript has been judged scientifically suitable for publication and will be formally accepted for publication once it meets all outstanding technical requirements.

Kind regards,

Laura Scherer

Academic Editor

PLOS ONE

---

## [Editor Report · Acceptance letter]

22 Nov 2021

PONE-D-20-40043R4 

Advancing the representation of reservoir hydropower in energy systems modelling: the case of Zambesi River Basin 

Dear Dr. Stevanato:

I'm pleased to inform you that your manuscript has been deemed suitable for publication in PLOS ONE. Congratulations! Your manuscript is now with our production department. 

Kind regards, 

on behalf of

Dr. Laura Scherer 

Academic Editor

PLOS ONE